# MaskCO: Masked Generation Drives Effective Representation Learning and Exploiting for Combinatorial Optimization

**Lvda Chen**[1†]**, Yang Li**[12†]**, Junchi Yan**[12] [*]

[1]School of AI & School of CS & Zhiyuan College, Shanghai Jiao Tong University
[2]Shanghai Innovation Institute
{chenlvda,yanglily,yanjunchi}@sjtu.edu.cn
https://github.com/Thinklab-SJTU/MaskCO

## Abstract

Neural Combinatorial Optimization (NCO) has long been anchored in paradigms such as solution construction or improvement that treat the solution as a monolithic reference, squandering the rich local decision patterns embedded in high-quality solutions. Inspired by the scalability of self-supervised pretraining in language and vision, we propose a shift in perspective: *Can combinatorial optimization adopt a fundamental training paradigm to enable scalable representation learning?* We introduce MaskCO, a masked generation approach that reframes learning to optimize as self-supervised learning on given reference solutions. By strategically masking portions of optimal solutions and training models to recover the missing content, MaskCO turns a single instance-solution pair into a multitude of local learning signals, forcing the model to internalize fine-grained structural dependencies. At inference time, we employ a mask-and-reconstruct procedure, i.e., a refinement loop that iteratively masks variables and regenerates them to progressively improve solution quality. Our findings show that these learned representations are highly transferable, facilitating effective fine-tuning and boosting the performance of alternative inference approaches. Experimental results demonstrate that MaskCO achieves remarkable performance improvements over previous state-of-the-art neural solvers, reducing the optimality gap by more than 99% and achieving a 10x speedup on problems such as the Travelling Salesman Problem (TSP).

## 1 Introduction

Combinatorial Optimization (CO), which seeks optimal solutions in discrete spaces under complex constraints, underpins numerous critical applications (Korte et al., 2011; Wu et al., 2024b;a; Yang et al., 2024). These problems, often NP-hard, pose significant challenges due to their inherent computational difficulty. Traditional approaches rely on hand-crafted heuristics that struggle to balance solution quality with computational efficiency (Helsgaun, 2017; Vidal et al., 2012). Recently, Neural Combinatorial Optimization (NCO) (Bengio et al., 2021; Cappart et al., 2021), has emerged as a transformative alternative, leveraging deep learning to automate heuristic design in a data-driven manner. Unlike traditional heuristics, NCO methods leverage structured distributions of problem instances to extract patterns directly from data or learn from objective feedback via customized algorithms, reducing reliance on manual intervention while achieving competitive or superior solution quality and computational efficiency (Kool et al., 2018; Sun & Yang, 2023; Li et al., 2024).

Many CO tasks can be cast as variable-decision problems on graphs, where the model seeks to identify a set of target variables that constitute a high-quality solution. Recent advancements often address these problems by relaxing discrete variables into continuous spaces and integrating objective-driven strategies. Leading approaches in this vein include unsupervised learning (Sanokowski et al., 2024),

[*]Correspondence author. † denotes equal contribution. This work was partly supported by NSFC (92370201, 625B2119), and Fundamental and Interdisciplinary Disciplines Breakthrough Plan of the Ministry of Education of China (JYB2025XDXM411).

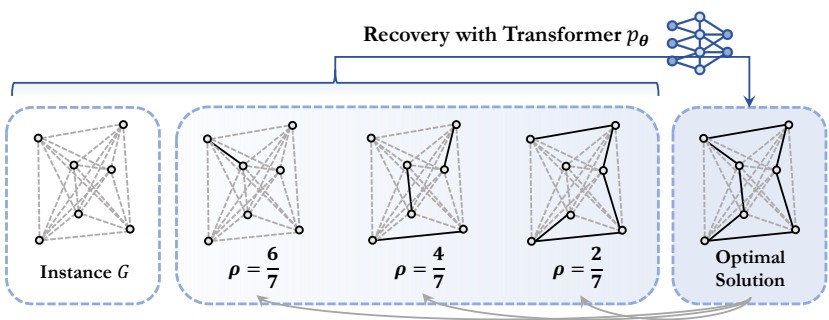

Figure 1: Overview of the training process, where MaskCO masks portions of optimal solutions and learns to recover the missing content. The masking ratio $\rho$ is uniformly sampled from [0,1].

objective-based fine-tuning (Sanokowski et al., 2023), and inference-time guidance (Li et al., 2023c; 2024; 2025a), largely grounded in diffusion models (Ho et al., 2020; Sun & Yang, 2023) or tree-based search algorithms (Fu et al., 2021). These approaches excel at optimizing soft-constrained forms for problems with relatively straightforward constraints, such as general node-decision problems on graphs like Maximum Independent Set (MIS) and edge-decision problems like Traveling Salesman Problem (TSP). However, they often struggle with more complex, hard-constrained scenarios like the Capacitated Vehicle Routing Problem (CVRP). Specifically for VRPs, approaches resort to sequential decision processes that enforce constraints at every step, either via autoregressive construction (Kool et al., 2018; Kwon et al., 2020; Kim et al., 2022; Luo et al., 2024; Zhou et al., 2024a) or via local-search actions formulated in a Markov Decision Process (MDP) and trained with Reinforcement Learning (RL) (da Costa et al., 2020; Sui et al., 2021; Ma et al., 2021; 2023).

As learning-based CO advances, model and training complexity have grown in tandem, with increasing reliance on problem-specific designs. Fundamentally, most existing approaches remain anchored in construction or improvement paradigms that treat a solution as a monolithic object. This instance-level focus leaves the rich localized decision patterns inherent in high-quality solutions largely unexplored, resulting in inefficient data utilization and limited scalability. This limitation is especially critical in CO, where obtaining reliable supervisory signals is often costly. This inefficiency mirrors historical challenges in natural language processing and computer vision, where progress was once constrained by a heavy reliance on task-specific labels. The field was revolutionized by self-supervised pretraining (Liu et al., 2021), driven by paradigms such as next-token prediction in GPT (Radford et al., 2018; Brown et al., 2020) and masked auto-encoding in BERT and MAE (Devlin et al., 2019; He et al., 2022). By enabling models to dive more deeply into raw data, these approaches unlock the ability to capture latent patterns at scale, achieving unprecedented generalization with minimal task-specific engineering. These successes invite a central question for CO: *Can we adopt a similarly foundational training paradigm that enables effective and scalable representation learning?*

In this paper, we propose a masked generation paradigm that redefines learning to optimize as self-supervised learning on given reference solutions. Our key insight is that optimal solutions, much like sentences or images, encode hierarchical decision patterns and recurring local substructures (e.g., efficient subroutes in routing, coherent node clusters in graph problems). By strategically masking portions of these solutions and training models to reconstruct the missing content, MaskCO compels the model to internalize fine-grained, localized decision logic. This approach exponentially increases data utility, transforming a single (instance, solution) pair into a diverse curriculum of (instance, partial-solution) training examples, as shown in Fig. 1.

At inference, we employ a multi-step parallel decoding that naturally leverages the training objective: at each step, the model simultaneously predicts all variables and selectively commits the highest-confidence components, repeating until the solution is complete. Furthermore, we introduce a mask-and-reconstruct procedure to enable a local-search-like refinement. By alternating between masking segments of a candidate solution and reconstructing them, the model progressively improves the solution quality over several iterations. The flexible masking mechanism affords fine-grained control over constraint satisfaction, bridging the gap between efficient heatmap-based inference and the handling of complex constraints. We also find that the learned representations transfer effectively to diverse inference pipelines and support efficient fine-tuning with alternative modeling formulations.

Our contributions are threefold: (1) We propose masked generation as a novel foundational paradigm for NCO, transforming solving into a solution-level self-supervised process. By training models to recover strategically masked components of optimal solutions, the framework forces the learning of fine-grained local decision patterns embedded in solution structures, achieving powerful representation learning while maintaining simplicity. (2) To fully exploit the learned representations, we design a direct inference algorithm that dynamically activates the models knowledge through iterative masking and regeneration, which mimics a computationally efficient search behavior. (3) Extensive experiments demonstrate the state-of-the-art performance of MaskCO on TSP, CVRP, and MIS.

## 2 RELATED WORKS

**Neural Combinatorial Optimization.** Learning-based combinatorial optimization (CO) solvers often utilize neural networks to either construct solutions or enhance existing ones, with the aim of directly minimizing objective scores (Bengio et al., 2021). This is typically achieved through reinforcement learning (Kool et al., 2018; Kwon et al., 2020; Kim et al., 2022; Qiu et al., 2022; Min et al., 2024) or by aligning predictions with reference solutions using supervised learning (Vinyals et al., 2015; Joshi et al., 2019; Hudson et al., 2022; Fu et al., 2021; Luo et al., 2024). ML4TSPBench (Li et al., 2025c) and ML4CO-Bench-101 (Ma et al., 2025b) provide systematic overviews and implementation foundations of the existing methods for the specific routing problem and the general CO domain.

Construction-based methods can be categorized into autoregressive and non-autoregressive approaches. Autoregressive methods (Khalil et al., 2017; Kool et al., 2018; Kwon et al., 2020; Hottung et al., 2021; Kim et al., 2022) sequentially determine decision variables until a complete solution is constructed, while non-autoregressive methods (Joshi et al., 2019; Fu et al., 2021; Geisler et al., 2022; Qiu et al., 2022; Zheng et al., 2024; Wang et al., 2023) predict soft-constrained solutions in one step, followed by post-processing to ensure feasibility. In recent years, generative models (Sun & Yang, 2023; Guo et al., 2024; Chen et al., 2024; Ma et al., 2025a; Wang et al.; Li et al., 2023c;b; 2024; 2025a;d) have shown promise in CO due to their strong representational power and the ability to model informative distributions, which is a non-autoregressive method with a higher model expressiveness. Despite their success, these methods typically operate at the monolithic instance-solution level, enforcing objectives only on the final output. This overlooks the granular, localized decision-making patterns inherent in optimal solutions. Conversely, improvement-based solvers (d O Costa et al., 2020; Wu et al., 2021; Chen & Tian, 2019; Li et al., 2021; Hou et al., 2023; Wang et al., 2025a) focus on refining solutions through local search guided by neural networks. While this allows for feedback during optimization, the lack of global guidance often renders such feedback less reliable. Furthermore, problem decomposition strategies (Luo et al., 2024; Drakulic et al., 2023; Ye et al., 2024) have explored decomposing large problems into smaller subproblems that can be locally solved and then integrated to achieve a global solution. While these strategies capture local solution patterns, they typically require specific pipeline designs tailored to particular problems.

More recently, inspired by the success of multi-task learning in CV and NLP, multi-task learning in CO has been studied (Drakulic et al., 2025; Zhou et al., 2024b; Berto et al., 2025; Pan et al., 2025; Zong et al., 2025). While some recent work has introduced self-supervision to NCO (Zong et al., 2025), MaskCO takes a distinct path. Rather than focusing solely on cross-problem generalization, we utilize self-supervised masked generation to maximize the representational capacity for specific problems. By internalizing localized structural dependencies, MaskCO achieves a new state-of-the-art in solution quality, providing a more robust architectural foundation for future multi-task extensions.

**Pre-training and Masked Generation.** Self-supervised learning techniques, which include pre-training on corrupted inputs or missing information, have gained considerable attention for their ability to learn useful representations without labeled data, fostering the development of foundation models (Li et al., 2025b; Long et al., 2026; Qin et al., 2025; Liu et al., 2025; 2026; Wang et al., 2025b). Masked generation (autoencoding), first introduced by BERT (Devlin et al., 2019) for natural language understanding, is a key concept in self-supervised pre-training. This method involves masking parts of the input and training models to predict the missing content, an approach that has proven highly effective across various domains. In the context of vision representation learning, this idea has been extended through the use of discrete tokenizers (Bao et al., 2022; He et al., 2021), demonstrating its ability to work effectively with non-sequentially structured data. Recent works (Chang et al., 2022; Li et al., 2023a) further demonstrate that masked generation can efficiently perform image synthesis in a

fixed number of steps using non-autoregressive decoding, showcasing its remarkable scalability and potential for broader applications, especially for the domain of CO, where problems are characterized by discrete decision spaces, exhibit a non-sequential nature, and require highly scalable methods.

## 3 PRELIMINARIES AND NOTATIONS

**Combinatorial Optimization on Graphs.** Following standard practice Karalias & Loukas (2020); Wang et al. (2022), we consider a family of graph instances $\mathcal{G}$. Each instance is a graph $G = (V, E)$ with vertex set $V$ and edge set $E$. Many CO tasks can be framed as selecting a subset of a ground set $U(G)$: edge-decision problems (e.g., TSP) select edges, so $U(G) = E$; node-decision problems (e.g., MIS) select vertices, so $U(G) = V$.

We represent a selection by a binary indicator $\mathbf{x} \in \{0,1\}^{|U(G)|}$, where $\mathbf{x}_u = 1$ means $u \in U(G)$ is selected. The feasible set $\Omega(G) \subseteq \{0,1\}^{|U(G)|}$ contains selections that satisfy the problems constraints. The goal is to find a feasible selection that minimizes a given objective $l(\cdot; G) : \{0,1\}^{|U(G)|} \to \mathbb{R}_{\geq 0}$:

$$\min_{\mathbf{x} \in \Omega(G)} l(\mathbf{x}; G). \tag{1}$$

Constraints can be enforced either as hard feasibility (encoded in $\Omega(G)$) or as soft penalties within $l(\cdot; G)$, depending on the problem. The selection problem can be defined below.

**Definition 3.1** (Selection Problem). A binary vector $\widehat{\mathbf{x}} \in \{0,1\}^{|U(G)|}$ is a *partial solution* for $G$ if there exists a feasible solution $\mathbf{x}' \in \Omega(G)$ such that $\widehat{\mathbf{x}} \leq \mathbf{x}'$ (elementwise). We denote the set of all partial solutions by $\widehat{\Omega}(G)$.

Intuitively, a partial solution selects a subset of variables that is consistent with at least one feasible full solution.

**Definition 3.2** (Selection Function). Given an instance $G$, a *selection operator* is a function

$$f_G : \widehat{\Omega}(G) \times [0,1]^{|U(G)|} \times \mathbb{N} \to \widehat{\Omega}(G) \tag{2}$$

that, given a partial solution $\widehat{\mathbf{x}}$, selection scores $\mathbf{p}$, and target size $k$, returns an extended partial solution satisfying:

- **Monotonicity:** $\widehat{\mathbf{x}} \leq f_G(\widehat{\mathbf{x}}, \mathbf{p}, k)$,
- **Target size or maximality:** either $|f_G(\widehat{\mathbf{x}}, \mathbf{p}, k)| \geq k$, or no element can be added without violating extendability (i.e., the result is maximal in $\widehat{\Omega}(G)$).

A typical instantiation is a greedy operator: sort $u \in U(G)$ by $\mathbf{p}_u$ and add elements one by one to $\widehat{\mathbf{x}}$ when the partial solution remains extendable; stop when the target size $k$ is reached or no further valid additions exist. With a fixed solution size, setting $k$ to that size yields a complete solution.

**Examples.** TSP is defined on a complete undirected graph where vertices are cities and edges carry non-negative weights (e.g., distances). The solution selects exactly $|V|$ edges forming a Hamiltonian cycle and minimizes the tour length. MIS is defined on an undirected graph $G = (V, E)$; the solution selects a maximum-cardinality subset of vertices with no adjacent pairs. For routing problems such as CVRP, capacity constraints can be treated as hard feasibility in $\Omega(G)$ or relaxed and embedded as penalties in $l(\cdot; G)$, depending on the solver design.

## 4 METHODOLOGY

We propose a general and principled paradigm for combinatorial optimization based on mask generation, a framework designed to be both flexible, accommodating diverse problem formulations, and minimalist in its reliance on problem-specific components.

### 4.1 SOLUTION-LEVEL SELF-SUPERVISED LEARNING IN TRAINING

Drawing inspiration from the success of masked auto-encoders in language and vision pre-training (He et al., 2022), where models learn robust representations by reconstructing corrupted inputs, we propose

a similar foundational paradigm for CO that enables scalable, efficient learning of decision patterns within optimal solutions. In CO, optimal solutions inherently encode recurring local substructures. By treating these solutions as composite graphs of substructures, we design a training task that requires the model to infer missing components based on partial contexts. This approach leverages the compositional nature of CO solutions, encouraging the model to learn reusable decision patterns rather than memorizing monolithic solutions.

Specifically, given a complete optimal solution $\mathbf{x}^\star \in \Omega(G)$, we construct a partial solution $\widehat{\mathbf{x}} \in \widehat{\Omega}(G)$ by sampling a subset of elements from $\mathbf{x}$. This subset sampling is governed by a distribution $\mathcal{D}_t$ over the interval $[0, 1]$, which controls the proportion of elements retained. In each training iteration, we first sample a ratio $t \sim \mathcal{D}_t$, and then uniformly retain $\lceil t \cdot |\mathbf{x}^\star| \rceil$ elements from $\mathbf{x}$ to form the partial solution $\widehat{\mathbf{x}}$. By default, we use a uniform distribution for $\mathcal{D}_t$, which is a straightforward strategy that masks an equal number of variables each time. From a single solution $\mathbf{x}^\star$, the sampling process generates exponentially many partial solutions, significantly improving data efficiency.

Let $\mathbf{p}_\theta(G, \widehat{\mathbf{x}}) \in [0, 1]^{|U(G)|}$ denote the models predicted selection probabilities conditioned on the instance $G$ and the partial solution $\widehat{\mathbf{x}}$. Using the ground-truth indicator $\mathbf{x}^\star$ as the target, we define the training loss as a variable-wise binary cross-entropy:

$$\mathcal{L}(\theta; G, \widehat{\mathbf{x}}, \mathbf{x}^\star) = \text{BCE}\big(\mathbf{p}_\theta(G, \widehat{\mathbf{x}}), \mathbf{x}^\star\big) = -\sum_{u \in U(G)} \big[\mathbf{x}_u^\star \log \mathbf{p}_{\theta,u} + (1 - \mathbf{x}_u^\star) \log(1 - \mathbf{p}_{\theta,u})\big]. \quad (3)$$

This formulation treats each masked position as an independent binary classification task, where the model learns to predict whether a variable belongs to the optimal solution based on the current context. Importantly, by iteratively masking different subsets of the solution during training, the model is exposed to diverse reasoning paths and learns a contextualized confidence estimator that generalizes well on diverse partial states.

## 4.2 MULTI-STEP DECODING FOR MASKED GENERATION

Our multi-step decoding framework progressively constructs solutions through an iterative refinement process that dynamically balances exploration and exploitation. The method leverages a schedule function $\gamma : [0, 1] \to [0, 1]$ to control the solution growth rate, which is monotonically increasing with $\gamma(0) = 0$ and $\gamma(1) = 1$. This function orchestrates a gradual transition from broad exploration of potential solution components to precise exploitation of the most promising elements. In practice, we adopt a linear implementation $\gamma(t) = t$, which provides uniform growth throughout the process.

For CO problems where all feasible solutions share a fixed cardinality $m$ (such as TSP tours with exactly $m$ edges), the decoding process proceeds through a predetermined sequence of $K$ deterministic steps. Beginning with an empty partial solution $\widehat{\mathbf{x}}^{(0)} = \mathbf{0}$, in each step $i \in [K]$ we adopt the greedy selection function that expands the solution by selecting the candidate variable with the highest predicted confidence according to the model's output heatmap $\mathbf{p}_\theta(G, \widehat{\mathbf{x}}^{(i-1)}) \in [0, 1]^{|U(G)|}$, among those not yet included. With selection function $f$, the incremental construction is well governed that the total number of selected elements reaches $\lceil \gamma(i/K) \cdot m \rceil$ at each step. Formally, we have:

$$\widehat{\mathbf{x}}^{(i)} \leftarrow f_G\Big(\widehat{\mathbf{x}}^{(i-1)}, \mathbf{p}_\theta(G, \widehat{\mathbf{x}}^{(i-1)}), \lceil \gamma(i/K) \cdot m \rceil\Big), \quad i \in [K]. \quad (4)$$

By construction, the final partial solution satisfies $|\widehat{\mathbf{x}}^{(K)}| = m$ and is feasible.

When dealing with problems exhibiting variable solution cardinality (such as MIS), an estimate $m_\theta(\widehat{\mathbf{x}})$ of the solution size is first obtained by summing the model's predictions:

$$m_\theta(G, \widehat{\mathbf{x}}) := \sum_{u \in U(G)} \mathbf{p}_{\theta,u}(G, \widehat{\mathbf{x}}). \quad (5)$$

Base on this estimate, accordingly, the iterative correction process becomes:

$$\widehat{\mathbf{x}}^{(i)} \leftarrow f_G\Big(\widehat{\mathbf{x}}^{(i-1)}, \mathbf{p}_\theta(G, \widehat{\mathbf{x}}^{(i-1)}), \lceil \gamma(i/K) \cdot m_\theta(G, \widehat{\mathbf{x}}^{(i-1)}) \rceil\Big), \quad i \in [K]. \quad (6)$$

Since the final iterate may not be feasible, if $\widehat{\mathbf{x}}^{(K)} \notin \Omega(G)$ we apply a completion step:

$$\widehat{\mathbf{x}} \leftarrow f_G\Big(\widehat{\mathbf{x}}^{(K)}, \mathbf{p}_\theta(G, \widehat{\mathbf{x}}^{(K-1)}), |U(G)|\Big), \quad (7)$$

reusing the previous scores to avoid recomputation and maintain consistency. Alternative completion strategies are discussed in Appendix B.4.

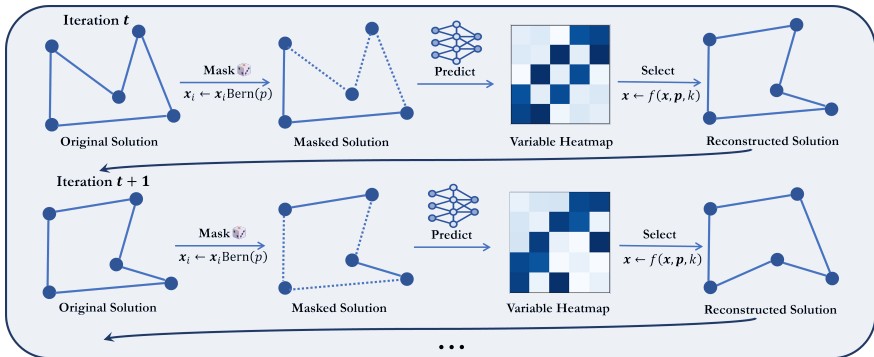

Figure 2: Overview of the iterative inference process. The solution is progressively refined by repeatedly masking and reconstructing parts of it based on the neural predictions.

### 4.3    THE HIGH-LEVEL CORRECTION FRAMEWORK: MASK AND RECONSTRUCT

The deterministic decoding process, while efficient, inherently limits exploration once an initial solution is formed, as it sequentially commits to irreversible decisions that may trap the construction in suboptimal configurations. To this end, we introduce a *mask-and-reconstruct* mechanism that enables iterative solution correction. Specifically, in each iteration, given a solution $\mathbf{x}$, a subset of its elements is randomly masked according to a predefined *keeping rate* $p$, resulting in a partial solution $\widehat{\mathbf{x}}$. During reconstruction, the model regenerates the masked regions through an adapted multi-step decoding process governed by a shifted schedule function $\gamma_p(r) = p + (1 - p) \cdot \gamma(r)$, which adjusts the decoding trajectory to focus on reconstructing the missing $1 - p$ fraction of the solution.

By anchoring the early steps of reconstruction to the retained elements in $\widehat{\mathbf{x}}$, the model leverages the unmasked portions as contextual anchors while probabilistically exploring alternative configurations for the masked regions. The shifted schedule $\gamma_p$ ensures that the decoding process begins from the partial solutions existing completeness level $p$ and gradually progresses toward full reconstruction over $K$ steps. Crucially, the masking rate $p$ controls the trade-off between exploration and exploitation: lower $p$ values lead to aggressive re-optimization of larger solution segments, while higher values of $p$ enable more localized, fine-grained improvements.

The full solving pipeline are presented in Fig. 2 and Alg. 1, consisting of two phases: *construction* and *correction*. In the construction phase,

---

**Algorithm 1** Inference Algorithm

1: **Input:**    multi-step    decoding    function $\text{Decode}_{f_G,\theta,K}(\cdot)$ with respect to instance $G$, model parameters $\theta$, selection operator $f_G$, and number of iterations per decode $K$; schedules $\gamma(\cdot)$ and $\gamma_p(\cdot)$; total budget $T$; masking rate $p \in [0, 1]$; objective $l(\cdot; G)$
2: $\mathbf{x} \leftarrow \text{Decode}_{\theta,f_G,K}(\mathbf{0}, \gamma)$
3: $\mathbf{x}_{\text{best}} \leftarrow \mathbf{x}$
4: **for** $i = 1$ to $\lfloor T/K \rfloor - 1$ **do**
5:     $\widehat{\mathbf{x}} \leftarrow \text{RandomlyMask}(\mathbf{x}, p)$
6:     $\mathbf{x} \leftarrow \text{Decode}_{\theta,f_G,K}(\widehat{\mathbf{x}}, \gamma_p)$
7:     **if** $l(\mathbf{x}; G) < l(\mathbf{x}_{\text{best}}; G)$ **then**
8:         $\mathbf{x}_{\text{best}} \leftarrow \mathbf{x}$
9:     **end if**
10: **end for**
11: **return** $\mathbf{x}_{\text{best}}$

---

an initial solution is generated from scratch using the base decoding method. This is followed by multiple rounds of the correction phase, in which the mask-and-reconstruct mechanism iteratively refines the solution, progressively enhancing its quality. This approach transforms static solution generation into an dynamic, memory-enhanced search process. Previously identified high-quality substructures serve as guides for subsequent refinements, while the controlled application of masking preserves the flexibility to explore novel regions of the solution space. Consequently, the model trained with the mask loss exhibits strong performance as an efficient combinatorial solver, demonstrating the effectiveness of transferring representation learning into practical problem-solving capabilities.

### 4.4    MODEL ARCHITECTURE

We adopt an encoder-decoder Transformer architecture (Vaswani, 2017), chosen for its capacity to model complex dependencies in combinatorial structures through self-attention mechanisms. Node features are linearly projected into the embedding space. Edge connections, represented by a 0-1 adjacency matrix, are incorporated into the attention mechanism through an additive bias. Formally,

Table 1: Results on TSP across various sizes.

| METHOD | TSP-100 | | | TSP-500 | | | TSP-1000 | | |
|---|---|---|---|---|---|---|---|---|---|
| | OBJ. | GAP | TIME | OBJ. | GAP | TIME | OBJ. | GAP | TIME |
| Concorde (Applegate et al., 2006) | 7.756 | 0.00% | 12m* | 16.55 | 0.00% | 37.66m* | 23.12 | 0.00% | 6.65h* |
| LKH-3 (Helsgaun, 2017) | 7.756 | 0.00% | 33m* | 16.55 | 0.00% | 46.28m* | 23.12 | 0.00% | 2.57h* |
| BQ-NCO (bs16) (Drakulic et al., 2023) | 7.757 | 0.016% | 1m41s | 16.637 | 0.551% | 5m50s | 23.436 | 1.374% | 14m30s |
| LEHD (RRC 1000) (Luo et al., 2024) | 7.756 | 0.0019% | 7.87m | 16.575 | 0.175% | 37.54m | 23.286 | 0.726% | 3.35h |
| SIT (RRC 1000) (Luo et al., 2025) | – | – | – | – | – | – | 23.206 | 0.381% | 1.59h |
| Poppy (16) (Grinsztajn et al., 2023) | 7.770* | 0.07%* | 1m* | – | – | – | – | – | – |
| PolyNet (sampling) (Hottung et al., 2025) | 7.756 | 0.0006% | 0.36m | – | – | – | – | – | – |
| PolyNet (EAS) (Hottung et al., 2025) | 7.756 | 0.0000% | 20.66m | – | – | – | – | – | – |
| DIFUSCO (T$_s$=100) (Sun & Yang, 2023) | 7.76 | 0.06% | 30m | 16.69 | 0.87% | 19.1m | 23.42 | 1.31% | 51.9m |
| Fast T2T (T$_s$=5,T$_g$=5) (Li et al., 2024) | 7.76 | 0.01% | 8.3m | 16.58 | 0.21% | 6.9m | 23.22 | 0.42% | 18.3m |
| MaskCO (T=320) | 7.756 | 0.0000% | 8s | 16.546 | 0.0020% | 6s | 23.120 | 0.0071% | 18s |
| MaskCO (T=640) | 7.756 | 0.0000% | 15s | 16.546 | 0.0014% | 12s | 23.119 | 0.0051% | 33s |
| MaskCO (T=1280) | 7.756 | 0.0000% | 30s | 16.546 | 0.0012% | 23s | 23.119 | 0.0038% | 1m6s |
| MaskCO (T=2560) | 7.756 | 0.0000% | 1m0s | 16.546 | 0.0007% | 43s | 23.119 | 0.0027% | 2m8s |

the attention logits are computed as $\mathbf{Z} = \mathbf{Q}^\top \mathbf{K} + \mathbf{A}$, where $\mathbf{Q}$ and $\mathbf{K}$ denote query and key matrices, respectively; $\mathbf{A}$ is the adjacency matrix encoding the presence or absence of edges between nodes, directly added to $\mathbf{Q}^\top \mathbf{K}$ without any scaling; and $\mathbf{Z}$ represents the resulting attention logits before applying the softmax function. For node-selection problems, node features are projected onto a one-dimensional space to obtain the logits. For edge-selection problems, pairwise compatibility is evaluated using the dot product of source and target node embeddings.

## 5 EXPERIMENTS

**Problem Settings.** We follow the standard data generation procedure to generate TSP, CVRP, and MIS datasets. For TSP, the $n$ node locations are sampled uniformly at random in the unit square (Kool et al., 2018; Kwon et al., 2020; Li et al., 2024; Zhou et al., 2024a; Ma et al., 2021; 2023). For CVRP, the depot location as well as $n$ customer locations are sampled uniformly at random in the unit square (Kool et al., 2018). The customer demands are sampled uniformly from $\{1, \cdots, 9\}$ and the vehicle capacity $D$ is set to 50 across problem sizes. For MIS, two datasets are tested, including RB graphs (Zhang et al., 2023) and ErdsRnyi (ER) graphs (Erdős et al., 1960). For RB graphs, we randomly sample 200 to 300 vertices uniformly and generate the graph instances. ER graphs are randomly generated with each edge maintaining a fixed probability of being present or absent, independently of the other edges. We adopt ER graphs of 700 to 800 nodes with the pairwise connection probability set as 0.15. Dataset configurations are provided in Appendix E.3.

**Solving Settings.** 1) Forward Passes $T$, i.e. the total number of model forward passes executed during the solving process for each instance. 2) Sampling Steps $K$, which parametrizes the multi-step decoding. 3) Keeping Rate $p$, i.e., the proportion of variables retained at each correction step. Consequently, the solving process consists of one initial construction phase followed by $T/K - 1$ correction iterations. The value of $T$ is reported in each table as the computational budget, while detailed configurations for $K$ and $p$ are provided in Appendix E.2. For VRPs, the 2-opt heuristic with penalty terms is applied between correction steps to enforce constraints and enhance solution quality. For all baselines, unless explicitly stated otherwise, we adopt the best-reported hyperparameters from the original papers. Metrics marked with an $^*$ are quoted from (Li et al., 2024; Grinsztajn et al., 2023). SIT on CVRP-1000 was retrained by us, as no publicly available checkpoints existed for the setting of $C = 50$. All other baselines are evaluated by assessing their provided checkpoints.

### 5.1 MAIN RESULTS

**TSP Results.** Table 1 presents the results of solvers on the TSP across different problem sizes. The performance of classical solvers is taken from Li et al. (2024). Traditional solvers like Concorde and LKH-3 achieve a 0.00% gap across all sizes, but their computation times are significantly longer, especially for larger problem sizes. In contrast, generative-based methods like DIFUSCO and Fast T2T show slightly higher gaps (0.06% and 0.01% for TSP-100), but they drastically reduce computation time, with Fast T2T completing TSP-100 in just 8.3 minutes. Notably, MaskCO outperforms all other methods, achieving near-zero gaps across all sizes while significantly reducing computation times. Compared to previous state-of-the-art neural solvers, MaskCO achieves remarkable performance improvements exceeding 99% in optimality gap reduction, along with a 10x speedup.

Table 2: Results on CVRP across various sizes.

| METHOD | CVRP-100 | | | CVRP-500 | | | CVRP-1000 | | |
|---|---|---|---|---|---|---|---|---|---|
| | OBJ. | GAP | TIME | OBJ. | GAP | TIME | OBJ. | GAP | TIME |
| HGS (Vidal et al., 2012) | 15.550 | 0.000% | 13h | 62.15 | 0.000% | 55h | 121.07 | 0.000% | 197h |
| POMO (Kwon et al., 2020) | 15.750 | 1.287% | 8s | – | – | – | – | – | – |
| ICAM (Zhou et al., 2024a) | 15.859 | 1.985% | 5s | 63.28 | 1.813% | 28s | 123.02 | 1.610% | 1m56s |
| UDC (Zheng et al., 2025) | – | – | – | 65.43 | 5.278% | 5m56s | 127.07 | 4.954% | 4m25s |
| BQ-NCO (bs16) (Drakulic et al., 2023) | 15.794 | 1.572% | 2.74m | 63.53 | 2.219% | 5.86m | 123.56 | 2.059% | 7.73m |
| LEHD (RRC 1000) (Luo et al., 2024) | 15.617 | 0.433% | 9.53m | 62.94 | 1.275% | 38.27m | 123.33 | 1.859% | 1.83h |
| ReLD (augx8) (Huang et al., 2025) | 15.785 | 1.509% | 0.12m | 64.16 | 3.241% | 0.17m | 125.06 | 3.297% | 0.34m |
| SIT (RRC 1000) (Luo et al., 2025) | – | – | – | – | – | – | 126.55 | 4.528% | 1.23h |
| Poppy (32) (Grinsztajn et al., 2023) | 15.73* | 1.06%* | 5m* | – | – | – | – | – | – |
| PolyNet (sampling) (Hottung et al., 2025) | 15.627 | 0.496% | 0.46m | – | – | – | – | – | – |
| PolyNet (EAS) (Hottung et al., 2025) | 15.571 | 0.139% | 25.40m | – | – | – | – | – | – |
| NeuOpt (D2A=5,T=40k) (Ma et al., 2023) | 15.566 | 0.103% | 4h49m | – | – | – | – | – | – |
| MaskCO (T=640) | 15.586 | 0.232% | 32s | 62.66 | 0.813% | 21s | 122.03 | 0.798% | 44s |
| MaskCO (T=1280) | 15.577 | 0.176% | 1m3s | 62.59 | 0.714% | 40s | 121.85 | 0.644% | 1m7s |
| MaskCO (T=2560) | 15.571 | 0.135% | 2m5s | 62.53 | 0.608% | 1m18s | 121.69 | 0.514% | 2m2s |
| MaskCO (T=5120) | 15.567 | 0.111% | 4m9s | 62.47 | 0.514% | 2m35s | 121.63 | 0.460% | 3m51s |
| MaskCO (T=10240) | 15.563 | 0.086% | 8m17s | 62.43 | 0.448% | 5m8s | 121.60 | 0.438% | 7m33s |

Table 3: Results on MIS across various sizes.

| METHOD | RB-[200-300] | | | ER-[700-800] | | |
|---|---|---|---|---|---|---|
| | OBJ. | GAP | TIME | OBJ. | GAP | TIME |
| KaMIS (Lamm et al., 2016) | 20.10* | – | 1.4h | 44.87* | – | 52.1m |
| Gurobi (Gurobi Optimization, 2020) | 19.98 | 0.01% | 47.6m | 41.28 | 7.78% | 50.0m |
| Intel (Li et al., 2018) | 18.47 | 8.11% | 13.1m | 38.80 | 13.43% | 20.0m |
| DGL (Böther et al., 2022) | 17.36 | 13.61% | 12.8m | 37.26 | 16.96% | 22.7m |
| GFlowNets (Zhang et al., 2023) | 19.18 | 4.57% | 32s | 41.14 | 8.53% | 2.9m |
| DIFUSCO (Sun & Yang, 2023) | 19.13 | 4.79% | 20.5m | 39.12 | 12.81% | 21.7m |
| T2T (Li et al., 2023c) | 19.38 | 3.53% | 30.3m | 41.41 | 7.72% | 27.8m |
| Fast T2T (Li et al., 2024) | 19.49 | 2.89% | 4.7m | 40.68 | 9.34% | 1.5m |
| MaskCO (T=1k) | 20.00 | 0.49% | 44s | 43.20 | 3.73% | 1m21s |
| MaskCO (T=2k) | 20.03 | 0.37% | 1m27s | 43.59 | 2.84% | 2m42s |
| MaskCO (T=4k) | 20.06 | 0.21% | 2m54s | 43.86 | 2.25% | 5m24s |
| MaskCO (T=8k) | 20.07 | 0.15% | 5m48s | 44.12 | 1.68% | 10m48s |
| MaskCO (T=16k) | 20.08 | 0.12% | 11m36s | 44.38 | 1.09% | 21m36s |
| MaskCO (T=32k) | 20.08 | 0.10% | 23m12s | 44.59 | 0.62% | 43m12s |

Table 4: Generalization on TSPLIB.

| | ICAM | Fast T2T | NeuOpt | MaskCO |
|---|---|---|---|---|
| TSPLIB50-200 | 2.38% | 0.28% | 0.35% | **0.033%** |
| TSPLIB201-1000 | 6.49% | 0.93% | – | **0.115%** |

Table 5: Ablation studies on the correction mechanism.

| CONFIG. | TSP-500 | | |
|---|---|---|---|
| | OBJ. | GAP | TIME |
| w/o corr. (T=K=1) | 16.689 | 0.8656% | <1s |
| w/o corr. (T=K=8) | 16.563 | 0.1015% | <1s |
| w/o corr. (T=K=64) | 16.556 | 0.0606% | 1s |
| w/o corr. (T=K=320) | 16.554 | 0.0520% | 6s |
| w/ corr. (T=320, K=1) | 16.546 | 0.0020% | 6s |

Table 4 presents generalization performance on the real-world TSPLIB dataset. We evaluate our model trained with 100-node problems on TSPLIB instances with 50-200 nodes and evaluate the 500-node model on TSPLIB with 200-1000 nodes using $T = 25600$. Compared to previous SOTAs, MaskCO achieves significant improvements from the 0.28% to 0.033% (88.2% improvement) on TSPLIB 50-200, and from 0.93% to 0.115% (87.6% improvement) on TSPLIB 200-1000, respectively.

**CVRP Results.** Table 2 highlights clear performance advantages of MaskCO. For CVRP-100, MaskCO (T=640) requires more time compared to constructive solvers (from 8s to 32s), but achieves a substantial improvement in solution quality (from 1.287% to 0.232%). Compared to search-based baselines like NeuOpt (Ma et al., 2023), although it achieves around 0.1% gap, it requires much more time (4h49m), whereas MaskCO can achieve 0.086% in just 8m17s, providing a 34.9x speedup. On larger instances, many baselines become infeasible. Compared to the SOTAs, MaskCO with 640 forward passes already outperforms previous methods in solution quality and solving speed, achieving an average performance gain of 53% along with a 64% speedup. By utilizing more computation, MaskCO can achieve remarkable optimality gaps of 0.448% and 0.438% on CVRP-500 and 1000, respectively. Moreover, MaskCO continues to improve with an increasing number of forward passes.

**MIS Results.** Table 3 shows that MaskCO with only 1,000 forward passes already outperforms previous state-of-the-art methods, achieving a 0.49% gap in 44 seconds on the RB dataset and a 3.73% gap in 1.3 minutes on the ER dataset. This results in an average 72% improvement in optimality gap and a 3x speedup in runtime compared to prior methods. As the number of forward passes increases, MaskCO continues to improve solution quality, consistently reducing the gap. With 32k forward passes, MaskCO achieves gaps of just 0.10% and 0.62% on the RB and ER datasets, respectively, setting new benchmarks by outperforming the best generative state-of-the-art methods by 94%.

## 5.2 ADAPTATIONS FOR ALTERNATIVE TRAINING AND DECODING METHODS

We demonstrate that models trained with masked signal modeling can be directly adapted to alternative decoding routines or through few-shot fine-tuning. Specifically, we assess the model using auto-regressive (AR) decoding, bidirectional AR decoding, relaxed AR decoding, and finetuning

Table 6: Results of fine-tuning and alternative decoding. S: number of samples; $T_s$: sampling steps.

| DECODING METHOD | TSP-100 | | | TSP-500 | | | TSP-1000 | | |
|---|---|---|---|---|---|---|---|---|---|
| | OBJ. | GAP | TIME | OBJ. | GAP | TIME | OBJ. | GAP | TIME |
| Direct Evaluation | | | | | | | | | |
| AR (aug×8) | 7.756 | 0.0004% | 19s | 16.549 | 0.018% | 1m42s | 23.136 | 0.078% | 10m4s |
| AR (2Opt, aug×8) | 7.756 | 0.0004% | 19s | 16.548 | 0.012% | 1m42s | 23.125 | 0.030% | 10m4s |
| Bidirectional AR (aug×8) | 7.756 | 0.0007% | 19s | 16.549 | 0.021% | 1m42s | 23.215 | 0.419% | 10m4s |
| Bidirectional AR (2Opt, aug×8) | 7.756 | 0.0006% | 19s | 16.549 | 0.019% | 1m42s | 23.141 | 0.098% | 10m4s |
| Relaxed AR (aug×8) | 7.756 | 0.0016% | 19s | 16.549 | 0.019% | 1m41s | 23.128 | 0.043% | 10m4s |
| Relaxed AR (2Opt, aug×8) | 7.756 | 0.0016% | 19s | 16.548 | 0.014% | 1m41s | 23.125 | 0.028% | 10m4s |
| 1-Epoch Finetuning | | | | | | | | | |
| Consistency (S=8, $T_s$=64) (Li et al., 2024) | 7.756 | 0.0008% | 15s | 16.552 | 0.040% | 29s | 23.148 | 0.131% | 2m13s |
| Consistency (S=8, $T_s$=256) (Li et al., 2024) | 7.756 | 0.0004% | 1m0s | 16.550 | 0.025% | 1m47s | 23.143 | 0.107% | 8m54s |

Table 7: Ablation studies on architecture.

| $T$ | BACKBONE | TSP-500 | TSP-1000 |
|---|---|---|---|
| 320 | Transformer | 0.0020%, 6s | 0.0071%, 18s |
| | GCN | 0.0212%, 40s | 0.0457%, 2m24s |
| 640 | Transformer | 0.0014%, 12s | 0.0051%, 33s |
| | GCN | 0.0141%, 1m15s | 0.0324%, 4m39s |
| 1280 | Transformer | 0.0012%, 23s | 0.0038%, 1m6s |
| | GCN | 0.0107%, 2m26s | 0.0229%, 9m19s |

Table 8: Ablation studies on mask modeling.

| METHOD | TSP-500 | TSP-1000 |
|---|---|---|
| DIFUSCO | 0.87%, 19.1m | 1.31%, 51.9m |
| Fast T2T | 0.21%, 6.9m | 0.42%, 18.3m |
| MaskCO-GCN (T=320) | 0.0212%, 40s | 0.046%, 2m24s |
| MaskCO-GCN (T=640) | 0.0141%, 1m15s | 0.032%, 4m39s |
| MaskCO-GCN (T=1280) | 0.0107%, 2m26s | 0.023%, 9m19s |

with consistency (Li et al., 2024). The bidirectional AR is specified for TSP, allowing the current tour segment to be extended on both ends. AR and bidirectional AR are implemented by applying an auto-regressive masking scheme on the output heatmap, ensuring that only edges associated with the end nodes of the partial tour can be selected at each step. For relaxed AR decoding, the requirement of maintaining a contiguous tour segment is relaxed; instead, it globally inserts one edge per step. This can be implemented by simply setting $T$ and $K$ equal to the problem size. We also consider MaskCO as a pre-trained model and fine-tune it with one of the SOTAs, i.e., the optimization consistency model (Li et al., 2024). Since the input requirement is different and thus direct evaluation is not feasible, we inherit the model weight and perform 1-epoch finetuning using its original training method.

**Experimental Results.** Notably, models trained with MaskCO can directly perform AR decoding and achieves superior performance compared to previous SOTA supervised AR methods like Drakulic et al. (2023); Luo et al. (2024), even though the latter employ additional boosting search techniques like beam search. This highlights the effectiveness of masked learning in acquiring richer representations for combinatorial optimization. Experiments on consistency presents that 1-epoch finetuning unlocks decoding with distinct formulations, verifying the generality of the learned representations.

## 5.3 ABLATION STUDIES

**Ablation on Correction Mechanism.** As shown in Table 5, on TSP-500 we observe an over 20 reduction in gap with the introduction of the correction mechanism. Similar to prior diffusion solvers, merely increasing the number of model inferences within a single constructive pass can deliver notable gains, but the efficiency of these gains is limited. In contrast, increasing the number of corrections via a mask-and-reconstruct procedure better leverages the training objective and the models capabilities, yielding a markedly superior cost-benefit ratio.

**Ablation on Architecture and Mask Modeling.** To isolate architectural effects, we replace our transformer with a Graph Convolution Network (GCN) that incorporates edge features (Joshi et al., 2019), keeping their settings except the network depth match our Transformer's configurations. Table 7 shows significant improvement from transformer architecture. Note that transformer models require only about $1/5$ of the training time on TSP-500 compared to GCN. This efficiency enables extensive training and contributes to the superior performance of transformers, making them better suited for challenging representation learning tasks. More comparison of training resources is available in Appendix F.4. To isolate the impact of masked modeling, we compare MaskCO with a GCN backbone (MaskCO-GCN) against other GCN-based methods, where masked modeling is the sole difference. Table 8 demonstrates that MaskCO-GCN achieves a **9x** reduction in optimality gap along with an **8x** speedup compared to Fast T2T, or alternatively, a **18x** reduction in gap with approximately a **2x** speedup. This highlights the strength of the masked modeling.

Table 9: Optimal-solution-free training on TSP-100 and 500 (T=2560). D0 denotes training on 2-opt labels; D$x$ ($x \geq 1$) denotes self-training using pseudo-labels from D($x-1$).

| METHOD | TSP-100 | | TSP-500 | |
|---|---|---|---|---|
| | OBJ. | GAP | OBJ. | GAP |
| 2Opt (128 runs) | 7.905 | 1.920% | 17.703 | 6.994% |
| MaskCO (D0) | 7.756 | 0.001% | 16.606 | 0.362% |
| MaskCO (D1) | 7.756 | 0.000% | 16.561 | 0.089% |
| MaskCO (D2) | 7.756 | 0.000% | 16.556 | 0.059% |

Table 10: Optimal-solution-free training on TSP-1000 with T=2560. "Gen." indicates training on TSP-500 and testing on TSP-1000.

| METHOD | TSP-1000 | |
|---|---|---|
| | OBJ. | GAP |
| 2Opt (128 runs) | 25.052 | 8.366% |
| MaskCO (Gen. as D0) | 23.288 | 0.733% |
| MaskCO (D1) | 23.197 | 0.342% |
| MaskCO (D2) | 23.186 | 0.295% |

## 5.4 EXPERIMENTS ON OPTIMAL-SOLUTION-FREE TRAINING PARADIGM

Inspired by distillation techniques for diffusion models (Luhman & Luhman, 2021; Liu et al., 2022) and SIT (Luo et al., 2025), we present a two-stage training variant that requires no optimal solutions yet surpasses SOTAs. Stage 1 initializes the model from low-quality heuristic labels (2-opt with 128 restarts); if a pretrained model is available, this stage can be skipped. Stage 2 performs self-training: the model alternates between pseudo-labeling unlabeled instances and lightweight fine-tuning on these labels, progressively improving without ground-truth optima. This paradigm trades labeled optimal solutions for compute: Stage 1 is standard supervised learning on heuristic labels; Stage 2 is unsupervised self-training that leverages the models ability to improve upon its own pseudo-labels.

As shown in Table 9, MaskCO trained solely on 2-opt outputs distills useful signal and outperforms its teacher. On TSP-100, although 2-opt exhibits a 1.9% optimality gap, MaskCO reaches a 0.001% gap to optimal. This stage mirrors the main training setup, differing only in the data source. In each iteration, we pseudo-label 65,536 TSP-50, 4,096 TSP-500, and 2,048 TSP-1000 instances, and fine-tune with roughly 1/60 the gradient steps of a full run. With just one iteration, MaskCO exceeds Fast T2T across all scales, as shown in Tables 9 and 10).

## 6 CONCLUSION

This paper presents MaskCO, a masked generation paradigm that defines the learning process of neural combinatorial optimization as a solution-level self-supervised learning process to enable effective and scalable representation learning. The dynamic inference algorithm through iterative masking and regeneration further unlocks the learned representations to simulate an efficient search process for problem solving. Experimental results demonstrate significant improvements over existing state-of-the-art neural solvers, with remarkable reductions in optimality gaps and substantial speedups in solving problems like TSP, CVRP and MIS. MaskCO shows potential to pave the way for performance advances and pre-trained models in combinatorial optimization.

## ETHICS STATEMENT

This work adheres to the ICLR Code of Ethics and has been conducted with a commitment to responsible research practices. The study does not involve human subjects, sensitive personal data, or potentially harmful applications. We have made efforts to ensure transparency, reproducibility, and scientific integrity throughout our research process, including accurate reporting of methods and results. No conflicts of interest exist regarding funding sources or affiliations that could influence the work. We aim for our research to contribute positively to the machine learning community and society at large, promoting fairness, accessibility, and open scientific inquiry. Should any unforeseen broader impacts arise, we remain committed to addressing them responsibly.

## REPRODUCIBILITY STATEMENT

We are committed to upholding high standards of scientific excellence and transparency in accordance with the ICLR Code of Ethics. To ensure the reproducibility of our results, we have included a detailed description of our methodology (Section 4), model architectures (Subsection 4.4), hyperparameters (Appendix E.2), inference algorithm (Algorithm 1), and dataset configuration (Appendix E.3) in the main text and appendices. Code will be made publicly available upon acceptance.

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

APPENDIX

# A  THE USE OF LARGE LANGUAGE MODELS (LLMS)

We employed large language models strictly as copy-editing assistants to improve clarity, readability, and style across the manuscript. The models were used for tasks such as rephrasing sentences, checking grammar, harmonizing terminology, and smoothing the flow of prose. They did not generate technical content, analyses, results, or conclusions, and they did not modify the substance of our claims. All AI-assisted edits were reviewed and approved by the authors, and no confidential data were provided to the models.

# B  DISCUSSIONS ON CERTAIN DESIGN CHOICES

## B.1  ON THE ABSENSE OF A SPECIAL [MASK] TOKEN

In standard masked generative modeling (Devlin et al., 2019; Li et al., 2023a; Chang et al., 2022), a special token [MASK] is introduced to the vocabulary $\mathcal{V}$. We define the extended vocabulary as $\mathcal{V}_{\text{mask}} = \mathcal{V} \cup \{[\text{MASK}]\}$. Trainable embeddings are associated with each token in the extended vocabulary, including [MASK], to represent a much larger effective vocabulary within a limited embedding dimensionality (e.g., modeling a 100k-token vocabulary using 1k-dimensional embeddings).

However, in Combinatorial Optimization (CO), we primarily deal with binary decision variables where $\mathcal{V} = \{0, 1\}$. Assigning $d$-dimensional embeddings to these variables is often computationally redundant. This is because binary variables are already real-valued and correspond to a minimal vocabulary size ($|\mathcal{V}| = 2$). For example, in vehicle routing problems (VRPs) with $n$ nodes and embedding dimension $d$, there are approximately $n^2/2$ binary variables. Introducing a [MASK] token would break this scalar efficiency, necessitating a higher-dimensional embedding space to distinguish the mask from the binary states. To maintain architectural simplicity and computational efficiency, we forgo the [MASK] token, opting instead for a formulation that treats masked positions through the lens of selection.

## B.2  SELECTION VS. DECISION FORMULATIONS

A decision problem, in brief, involves assigning $N$ binary variables in order to minimize an objective function under certain constraints. Although it is theoretically equivalent to the selection problem formulation, we find that using the selection formulation offers several practical advantages:

- **Elimination of Mask Tokens:** By framing the task as selecting the next components to add to a partial solution, we naturally sidestep the need for an explicit [MASK] token, inheriting the efficiency discussed in Sec. B.1.

- **Intuitive Modeling:** The selection paradigm provides a more natural way to model many CO problems. For instance, in VRPs, the solution is naturally expressed as selecting edges to form a route, while in the MIS problem, the goal is to select a subset of nodes. Moreover, many existing methods that decode a heatmap into a solution essentially solve a selection problem. This approach, often referred to as *greedy insertion* in prior works (Sun & Yang, 2023; Li et al., 2023c; 2024), corresponds directly to the selection function formulation adopted in this paper.

## B.3  THE SELECTION FUNCTION

In this work, we employ a *greedy selection function*, where the model iteratively commits to the variables with the highest predicted confidence. We choose this straightforward strategy to isolate and demonstrate the inherent representational power of MaskCO, rather than relying on sophisticated selection functions. Nevertheless, the selection function remains a modular component. In more complex scenarios, it could be replaced by maximizing the average score of newly selected variables, which can be implemented using dynamic programming or bounded-width tree search.

### B.4 Alternative Strategies for Complete Solution Enforcement

Recall that in multi-step decoding for problems with variable solution cardinality, if $\widehat{\mathbf{x}}^{(K)} \notin \Omega(G)$ we apply a completion step:

$$\widehat{\mathbf{x}} \leftarrow f_G\Big(\widehat{\mathbf{x}}^{(K)}, \mathbf{p}_\theta(G, \widehat{\mathbf{x}}^{(K-1)}), |U(G)|\Big), \tag{8}$$

reusing the previous scores to avoid recomputation and maintain consistency. This strategy selects as many elements as possible, making it particularly suitable for problems that favor large solution cardinality, such as MIS. An alternative approach is to select only the minimal number of elements required to form a valid solution. This can be achieved by iteratively applying

$$\widehat{\mathbf{x}} \leftarrow f_G\Big(\widehat{\mathbf{x}}, \mathbf{p}_\theta(G, \widehat{\mathbf{x}}^{(K-1)}), |\mathrm{supp}(\widehat{\mathbf{x}})| + 1\Big), \tag{9}$$

until $\widehat{\mathbf{x}}$ becomes a complete solution.

## C  Comparison with Auto-Regressive Modeling

The basic decoding of MaskCO can be viewed as **a 2-step extension of the auto-regressive modeling** for VRPs:

**1. Select an edge for the current node → Select an edge for any node.** While autoregressive modeling traditionally selects an edge for the current node, i.e., the end node of the partial route, based on local edge confidence, we extend this approach to allow selecting an edge for any node, guided by global edge confidence. The flexibility of the non-sequential selection strategy ensures that globally confident edges are not overlooked simply because they conflict with previously selected edges, which may have been chosen due to high local confidence but lower global relevance. Thus, this extension mitigates the risk of suboptimal decisions caused by early, locally favorable choices that may hinder globally optimal solutions. Using only this extension corresponds to MaskCO with $K$ set equal to the number of nodes.

**2. Single-step prediction → Multi-step prediction.** While autoregressive modeling traditionally selects one edge from a single prediction, we extend it to select multiple edges from one prediction, where the number of edges is decoupled with training and can be adjusted in the inference stage. This extension provides flexibility of speed-quality trade-off and often leads to better scalability. Using only this extension resembles multi-token prediction in NLP.

During training, the partial solutions sampled in MaskCO are significantly richer than those used in traditional auto-regressive methods. While conventional auto-regressive approaches train only on contiguous segments of the solution, MaskCO trains on arbitrary subsets of the optimal solution, regardless of order or continuity. This exposes the model to a more diverse set of problem instances and partial solution structures, encouraging a deeper understanding of the underlying combinatorial problem. As a result, the model learns more robust and generalizable representations (especially when training data is limited), which translates into improved performance, even when using standard auto-regressive decoding at inference time.

For node-selection problems such as MIS, standard auto-regressive models already perform global node selection. In this case, for inference, MaskCO primarily introduces the second extension (i.e., multi-step prediction), which enhances efficiency and solution refinement without altering the global selection mechanism. In terms of training, conventional auto-regressive approaches require the model to predict nodes sequentially, one at a time. However, since solutions to node-selection problems are typically unordered, such sequential modeling imposes an artificial ordering, leading to ambiguous training signals. In contrast, MaskCO trains the model to predict the complete solution in a set-based manner, providing a more consistent and unambiguous training signal that aligns better with the non-sequential nature of the problem.

## D  Model Architecture Details

The model adopts a Transformer-based architecture, utilizing Pre-normalization (Pre-norm) and the SwiGLU (Shazeer, 2020) activation function.

Let $d_{\text{model}}$ denote the embedding dimension, $h$ the number of attention heads, $d_{\text{head}} = d_{\text{model}}/h$ the head dimension, and $L$ the sequence length. Given the input to a layer $\mathbf{X} \in \mathbb{R}^{L \times D_{\text{model}}}$ and the adjacency matrix $\mathbf{A} \in \{0,1\}^{L \times L}$, the layer output $\mathbf{Y}$ is computed through the following sublayers, following the pre-norm structure.

### D.1 MULTI-HEAD SELF-ATTENTION (MHSA)

The core attention mechanism, which includes the adjacency matrix $\mathbf{A}$ as a bias term, is defined as:

$$\text{Attention}(\mathbf{Q}, \mathbf{K}, \mathbf{V}, \mathbf{A}) = \text{Softmax}(\mathbf{Q}\mathbf{K}^\top + \mathbf{A})\mathbf{V} \tag{10}$$

The Multi-Head Self-Attention function $\text{MultiHeadSelfAttention}(\mathbf{X}, \mathbf{A})$ concatenates $h$ attention heads $\text{head}_i$ followed by an output projection $\mathbf{W}^O$:

$$\text{MultiHeadSelfAttention}(\mathbf{X}, \mathbf{A}) = \text{Concat}(\text{head}_1, \dots, \text{head}_h)\mathbf{W}^O, \tag{11}$$

where each head $\text{head}_i$ is computed as:

$$\text{head}_i = \text{Attention}(\mathbf{X}\mathbf{W}_i^Q, \mathbf{X}\mathbf{W}_i^K, \mathbf{X}\mathbf{W}_i^V, \mathbf{A}). \tag{12}$$

The weight matrices are dimensioned as $\mathbf{W}_i^Q, \mathbf{W}_i^K, \mathbf{W}_i^V \in \mathbb{R}^{d_{\text{model}} \times d_{\text{head}}}$ for the projection of Query, Key, and Value within head $i$, and the output projection is $\mathbf{W}^O \in \mathbb{R}^{d_{\text{model}} \times d_{\text{model}}}$.

### D.2 LAYER FORWARD PASS

The layer updates $\mathbf{X}$ sequentially through the MHSA sublayer and the Feed-Forward Network (FFN) sublayer, both employing residual connections:

#### D.2.1 SELF-ATTENTION SUBLAYER

The layer performs normalization before the attention operation (Pre-norm) and applies the residual connection:

$$\mathbf{X} \leftarrow \mathbf{X} + \text{MultiHeadSelfAttention}(\text{Norm}(\mathbf{X}), \mathbf{A}) \tag{13}$$

#### D.2.2 FEED-FORWARD SUBLAYER WITH SWIGLU

The layer again performs Pre-norm, applies the SwiGLU FFN, and adds a residual connection:

$$\mathbf{X} \leftarrow \mathbf{X} + \text{SwiGLU}(\text{Norm}(\mathbf{X})) \tag{14}$$

The final $\mathbf{X}$ is the output $\mathbf{Y}$ of the Transformer layer.

### D.3 INPUT AND OUTPUT LAYERS

**Input Layer.** For VRPs, raw node features are projected into model embedding space via a learnable linear layer. For MIS, node features are sampled from a standard normal distribution and linearly projected into the encoder embedding space.

**Output Layer.** For VRPs, the logit representing the connection between two different nodes $i$ and $j$ with transformed embedding $\mathbf{x}_i$ and $\mathbf{x}_j$ is computed via inner product $\langle \mathbf{x}_i, \mathbf{x}_j \rangle$. For MIS, the logit for selecting node $i$ is produced by projecting its embedding $\mathbf{x}_i$ to a single dimension.

## E EXPERIMENTAL DETAILS

### E.1 HARDWARE AND BASELINE SETTINGS

All experiments were conducted on a computing platform equipped with an NVIDIA A100 GPU and a 32-core Intel Xeon Platinum 8352S CPU. Traditional solver baselines (Concorde, LKH-3, HGS, KaMIS) were evaluated in single-threaded mode, following Ma et al. (2023); Sun & Yang (2023); Zhou et al. (2024a). For baselines without specified hyperparameters, we use the configuration yielding the best solution quality in their original papers. Regarding SIT (Luo et al., 2025) on CVRP-1000, since the publicly released checkpoint was not trained under the capacity setting $C = 50$, direct evaluation on our test set is not feasible. To ensure a fair comparison, we retrain their model from scratch on CVRP-1000 with $C = 50$, which requires approximately 10.7 days on a single A100 GPU.

Table 11: Neural network configurations.

|  | TSP | CVRP | MIS-RB-[200-300] | MIS-ER-[700-800] |
|---|---|---|---|---|
| embedding dimension | 256 | 512 | 256 | 256 |
| head dimension | 32 | 64 | 64 | 64 |
| encoder layers | 16 | 16 | 0 | 0 |
| decoder layers | 6 | 6 | 12 | 24 |

### E.2 HYPERPARAMETERS

#### E.2.1 NEURAL NETWORKS

The hyperparameters for the neural networks are detailed in Table 11. The encoder refers to the component that processes only the problem instance (i.e., the generation condition), while the decoder additionally processes the partial solution.

#### E.2.2 INFERENCE CONFIGURATION

The inference hyperparameters are specified in Tables 12, 13, and 14 for TSP, CVRP, and MIS, respectively.

Regarding the two key hyperparameters, $p$ and $K$, each plays a distinct role in guiding the inference process. The parameter $p$ explicitly controls the trade-off between exploitation and exploration: if $p$ is too large, the reconstructed solution remains overly close to the original, potentially causing the correction process to stagnate; if $p$ is too small, the reconstruction may fallback to generating solutions from scratch. Meanwhile, $K$ governs how forward passes are distributed—favoring either more sampling steps per iteration or more iterations with fewer steps. For simpler tasks, smaller values of $K$ are preferable, promoting more iterative refinement. In contrast, for harder tasks where predictions exhibit higher uncertainty (can be measured by entropy after normalizing prediction into probabilities), larger $K$ values are beneficial to gather more informative samples.

Beyond their semantic interpretations, several practical factors influence hyperparameter selection. Noisy data, for instance, increases prediction uncertainty and thus favors larger $K$. We verify this phenomenon by comparing the optimal hyperparameters for models trained on clean (optimal) data versus highly noisy data (generated using 128 runs of 2-opt) as shown in Table 18. The results show that noise shifts the optimal value of $K$ from 1 to 8. For large-scale CVRP, this phenomenon may also arise due to the stringent time limits imposed during data generation (4 minutes per instance for CVRP-500 and 8 minutes for CVRP-1000), resulting in noisy training data. Additionally, imbalanced learning across different values of $p$ can occur, i.e., even though $p$ is uniformly sampled during training, models often perform worse on medium-range $p$ values, as these correspond to more challenging reconstruction scenarios and may be underrepresented in effective learning. These compounding factors make identifying optimal hyperparameters non-trivial, even when their roles are well understood.

Fortunately, in our case, the extremely fast evaluation time of MaskCO enables efficient hyperparameter tuning via grid search. For example, full hyperparameter selection for TSP-500 takes only 3.6 minutes. Additional details and empirical analysis can be found in Appendix F.3. In the future works, $p$ may not be fixed as a constant; instead, it could be sampled from a distribution, set periodically, or even dynamically controlled by an auxiliary neural network.

Table 12: Inference configurations for TSP.

|  | TSP-100 | TSP-500 | TSP-1000 |
|---|---|---|---|
| Sampling Steps $K$ | 1 | 1 | 2 |
| Keeping Rate $p$ | 0.2 | 0.2 | 0.1 |

Table 13: Inference configurations for CVRP.

|  | CVRP-100 | CVRP-500 | CVRP-1000 |
|---|---|---|---|
| Sampling Steps $K$ | 2 | 16 | 128 |
| Keeping Rate $p$ | 0.3 | 0.3 | 0.6 |

Table 14: Inference configurations for MIS.

|  | RB-[200-300] | ER-[700-800] |
|---|---|---|
| Sampling Steps $K$ | 1 | 1 |
| Keeping Rate $p$ | 0.6 | 0.5 |

Table 15: Dataset Configurations for TSP.

|  | TSP-100 | TSP-500 | TSP-1000 |
|---|---|---|---|
| Training Dataset Size | 1,280K | 128K | |
| Training Dataset Solver | Concorde (Applegate et al., 2006) | LKH-3 (Helsgaun, 2017) | |
| Test Dataset Size | 1,280 | 128 | |
| Test Dataset Solver | Concorde (Applegate et al., 2006) | | |

Table 16: Dataset Configurations for CVRP.

|  | CVRP-100 | CVRP-500 | CVRP-1000 |
|---|---|---|---|
| Training Dataset Size | 1,536K | 200K | 100K |
| Training Dataset Solver | HGS (Vidal et al., 2012) | HGS with Decomposition (Santini et al., 2023) | |
| Test Dataset Size | 1,280 | 128 | 64 |
| Test Dataset Solver | HGS (Vidal et al., 2012) | | |

Table 17: Dataset Configurations for MIS.

|  | RB-[200-300] | ER-[700-800] |
|---|---|---|
| Training Dataset Size | 90,000 | 163,840 |
| Training Dataset Solver | KaMIS (Lamm et al., 2016) | |
| Test Dataset Size | 500 | 128 |
| Test Dataset Solver | KaMIS (Lamm et al., 2016) | |

### E.3 DATASETS

The dataset configurations for TSP, CVRP, and MIS are summarized in Tables 15, 16, and 17, respectively.

Table 18: Comparison of optimal hyperparameters for models trained on clean (optimal) data versus highly noisy data generated by 128-run 2-opt. For the model trained on clean data, the optimal configuration is $(K, p) = (1, 0.2)$, whereas for noisy data it is $(K, p) = (8, 0.2)$. Furthermore, for a fixed $K$, the optimal $p$ varies depending on data quality, indicating that prediction uncertainty influences hyperparameter selection in multiple dimensions.

| | | TSP-500 | |
|---|---|---|---|
| $K$ | $p$ | CLEAN DATA | NOISY DATA |
| | 0.1 | 0.0041% | 1.9819% |
| | 0.2 | **0.0020%** | 1.2726% |
| | 0.3 | 0.0051% | 0.9309% |
| | 0.4 | 0.0073% | 0.8800% |
| 1 | 0.5 | 0.0115% | 0.9427% |
| | 0.6 | 0.0179% | 1.2118% |
| | 0.7 | 0.0257% | 1.5670% |
| | 0.8 | 0.0443% | 1.9170% |
| | 0.9 | 0.0917% | 2.1689% |
| | 0.1 | 0.0048% | 0.6528% |
| | 0.2 | 0.0070% | **0.6371%** |
| | 0.3 | 0.0075% | 0.6710% |
| | 0.4 | 0.0093% | 0.7144% |
| 8 | 0.5 | 0.0095% | 0.7767% |
| | 0.6 | 0.0107% | 0.8466% |
| | 0.7 | 0.0111% | 0.8978% |
| | 0.8 | 0.0112% | 0.9291% |
| | 0.9 | 0.0113% | 0.9393% |

# F    SUPPLEMENTARY EXPERIMENTS

## F.1    INFLUENCE ON 2-OPT AND RELAXED SPACE FOR SOLVING VRPS

For the CVRP, we follow the practice of advanced traditional solvers such as HGS and LKH-3, which relax the capacity constraints to enable a larger search space and incorporate them into the cost function as penalty terms. Table 19 shows the effectiveness of the 2-opt heuristic (with penalty terms) in solving VRPs, particularly for CVRP under capacity relaxation, where it contributes significantly to effective constraint handling. In practice, full convergence of 2-opt is not required in every iteration. We find that $n/100$ and $n/25$ steps are typically sufficient for TSP and CVRP, respectively, where $n$ denotes the number of nodes.

We observed performance improvement by adopting the capacity-relaxed space. Results in Table 20 show that relaxation brings consistent improvement. Actually, advanced traditional solvers such as HGS and LKH-3 operate within a relaxed search space, and NeuOpt has demonstrated that exploring infeasible solutions can yield benefits for learning-based search methods. We believe that introducing 2-opt at test time is a relatively simple way to use the relaxed space, or some mechanisms like those used in NeuOpt should be included when training.

We also retain the flexibility to directly enforce a strictly feasible solution space when required by the experimental setup. In Table 21, we present the performance results of MaskCO under such a strictly feasible setting, where the selection function alone is used to handle constraints.

## F.2    GENERALIZATION STUDIES

### F.2.1    TSP

TSPLIB results have been demonstrated in 4. Cross-scale generation results is shown in Table 22.

Table 19: The impact of incorporating 2-opt heuristics in solving the TSP and CVRP. "− 2Opt" indicates that 2-opt is disabled between correction steps. "+ PostProcess." denotes that 2-opt is applied as a post-processing step after the final correction.

| METHOD | TSP-100 | | TSP-500 | | TSP-1000 | |
|---|---|---|---|---|---|---|
| | OBJ. | GAP | OBJ. | GAP | OBJ. | GAP |
| MaskCO (T=320) | 7.756 | 0.0000% | 16.546 | 0.0020% | 23.120 | 0.0071% |
| − 2Opt | 7.756 | 0.0000% | 16.546 | 0.0029% | 23.121 | 0.0110% |
| + PostProcess. | 7.756 | 0.0000% | 16.546 | 0.0029% | 23.120 | 0.0100% |
| MaskCO (T=640) | 7.756 | 0.0000% | 16.546 | 0.0014% | 23.119 | 0.0051% |
| − 2Opt | 7.756 | 0.0000% | 16.546 | 0.0017% | 23.120 | 0.0079% |
| + PostProcess. | 7.756 | 0.0000% | 16.546 | 0.0017% | 23.120 | 0.0075% |
| MaskCO (T=1280) | 7.756 | 0.0000% | 16.546 | 0.0012% | 23.119 | 0.0038% |
| − 2Opt | 7.756 | 0.0000% | 16.546 | 0.0013% | 23.119 | 0.0054% |
| + PostProcess. | 7.756 | 0.0000% | 16.546 | 0.0013% | 23.119 | 0.0054% |
| MaskCO (T=2560) | 7.756 | 0.0000% | 16.546 | 0.0007% | 23.119 | 0.0027% |
| − 2Opt | 7.756 | 0.0000% | 16.546 | 0.0008% | 23.119 | 0.0044% |
| + PostProcess. | 7.756 | 0.0000% | 16.546 | 0.0008% | 23.119 | 0.0044% |

| METHOD | CVRP-100 | | CVRP-500 | | CVRP-1000 | |
|---|---|---|---|---|---|---|
| | OBJ. | GAP | OBJ. | GAP | OBJ. | GAP |
| MaskCO (T=640) | 15.586 | 0.232% | 62.66 | 0.813% | 122.03 | 0.798% |
| − 2Opt | 15.675 | 0.808% | Inf | Inf% | Inf | Inf% |
| + PostProcess. | 15.610 | 0.384% | 62.698 | 0.882% | 122.16 | 0.898% |
| MaskCO (T=1280) | 15.577 | 0.176% | 62.59 | 0.714% | 121.85 | 0.644% |
| − 2Opt | 15.630 | 0.514% | Inf | Inf% | Inf | Inf% |
| + PostProcess. | 15.601 | 0.329% | 62.664 | 0.826% | 121.98 | 0.750% |
| MaskCO (T=2560) | 15.571 | 0.135% | 62.53 | 0.608% | 121.69 | 0.514% |
| − 2Opt | 15.606 | 0.360% | Inf | Inf% | Inf | Inf% |
| + PostProcess. | 15.593 | 0.277% | 62.642 | 0.791% | 121.81 | 0.615% |
| MaskCO (T=5120) | 15.567 | 0.111% | 62.47 | 0.514% | 121.63 | 0.460% |
| − 2Opt | 15.592 | 0.270% | Inf | Inf% | Inf | Inf% |
| + PostProcess. | 15.584 | 0.223% | 62.621 | 0.758% | 121.69 | 0.516% |
| MaskCO (T=10240) | 15.563 | 0.086% | 62.43 | 0.448% | 121.60 | 0.438% |
| − 2Opt | 15.582 | 0.205% | Inf | Inf% | Inf | Inf% |
| + PostProcess. | 15.579 | 0.186% | 62.620 | 0.756% | 121.72 | 0.537% |

Table 20: The impact of capacity relaxation on CVRP. "− Relaxation" indicates that the capacity relaxation is disabled.

| METHOD | CVRP-100 | | CVRP-500 | | CVRP-1000 | |
|---|---|---|---|---|---|---|
| | OBJ. | GAP | OBJ. | GAP | OBJ. | GAP |
| MaskCO (T=640) | 15.586 | 0.232% | 62.66 | 0.813% | 122.03 | 0.798% |
| − Relaxation | 15.661 | 0.717% | 64.60 | 3.944% | 126.36 | 4.370% |
| MaskCO (T=1280) | 15.577 | 0.176% | 62.59 | 0.714% | 121.85 | 0.644% |
| − Relaxation | 15.630 | 0.515% | 64.22 | 3.327% | 125.61 | 3.752% |
| MaskCO (T=2560) | 15.571 | 0.135% | 62.53 | 0.608% | 121.69 | 0.514% |
| − Relaxation | 15.608 | 0.374% | 63.87 | 2.768% | 124.82 | 3.095% |
| MaskCO (T=5120) | 15.567 | 0.111% | 62.47 | 0.514% | 121.63 | 0.460% |
| − Relaxation | 15.593 | 0.276% | 63.60 | 2.326% | 124.17 | 2.559% |
| MaskCO (T=10240) | 15.563 | 0.086% | 62.43 | 0.448% | 121.60 | 0.438% |
| − Relaxation | 15.583 | 0.212% | 63.32 | 1.885% | 123.91 | 2.347% |

Table 21: Results for using the selection function alone to handle constraints on CVRP. 2-opt and capacity relaxation are disabled.

| METHOD | CVRP-100 | | CVRP-500 | | CVRP-1000 | |
|---|---|---|---|---|---|---|
| | OBJ. | GAP | OBJ. | GAP | OBJ. | GAP |
| MaskCO (T=640) | 15.661 | 0.717% | 64.60 | 3.944% | 126.36 | 4.370% |
| MaskCO (T=1280) | 15.630 | 0.515% | 64.22 | 3.327% | 125.61 | 3.752% |
| MaskCO (T=2560) | 15.608 | 0.374% | 63.87 | 2.768% | 124.82 | 3.095% |
| MaskCO (T=5120) | 15.593 | 0.276% | 63.60 | 2.326% | 124.17 | 2.559% |
| MaskCO (T=10240) | 15.583 | 0.212% | 63.32 | 1.885% | 123.91 | 2.347% |

Table 22: Cross-scale generalization results for TSP.

| | Training / Testing | TSP-100 | TSP-500 | TSP-1000 |
|---|---|---|---|---|
| TSP100 | Fast T2T ($T_s$=20, $T_g$=20) | 7.76, 0.01% | **7.77, 0.23%** | 7.78, 0.34% |
| | MaskCO (T=2560) | **7.756, 0.0000%** | 7.796, 0.521% | 7.790, 0.437% |
| | MaskCO (T=10240) | **7.756, 0.0000%** | 7.773, 0.227% | **7.770, 0.179%** |
| TSP500 | Fast T2T ($T_s$=20, $T_g$=20) | 16.97, 2.54% | 16.58, 0.20% | 16.60, 0.33% |
| | MaskCO (T=2560) | 16.957, 2.485% | **16.546, 0.0007%** | 16.546, 0.0005% |
| | MaskCO (T=10240) | **16.903, 2.156%** | **16.546, 0.0007%** | **16.546, 0.0001%** |
| TSP1K | Fast T2T ($T_s$=20, $T_g$=20) | **24.01, 3.87%** | 23.25, 0.58% | 23.20, 0.36% |
| | MaskCO (T=2560) | 24.206, 4.707% | 23.134, 0.0702% | 23.119, 0.0027% |
| | MaskCO (T=10240) | 24.136, 4.404% | **23.129, 0.0456%** | **23.118, 0.0012%** |

Table 23: Generalization Results on VRPLIB (T=40960).

| | ICAM | NeuOpt | MaskCO |
|---|---|---|---|
| VRPLIB50-200 | 4.41% | 2.62% | **2.15%** |
| VRPLIB201-500 | 3.92% | – | **3.87%** |

Table 24: Generalization results for MIS. (T=32k)

| Training / Testing | RB-[200-300] | ER-[700-800] |
|---|---|---|
| RB-[200-300] | 20.08, 0.10% | 19.96, 0.72% |
| ER-[700-800] | 38.57, 14.04% | 44.59, 0.62% |

### F.2.2 CVRP

Results on the VRPLIB benchmark are summarized in Table 23. The mixed-scale and mixed-capacity training strategy proposed by ICAM (Zhou et al., 2024a) can be further incorporated into our method to enhance its generalization performance.

### F.2.3 MIS

The model is trained on RB-[200-300] and evaluated on ER-[700-800], and vice versa, as shown in Table 24.

## F.3 HYPERPARAMETER STUDIES

For the inference hyperparameters, we conduct a grid search over the number of sampling steps $K$ and the keeping rate $p$, while keeping the total number of forward passes fixed. The results are shown in Figures 3, 4, 5, 6, 7, 8, 9, and 10.

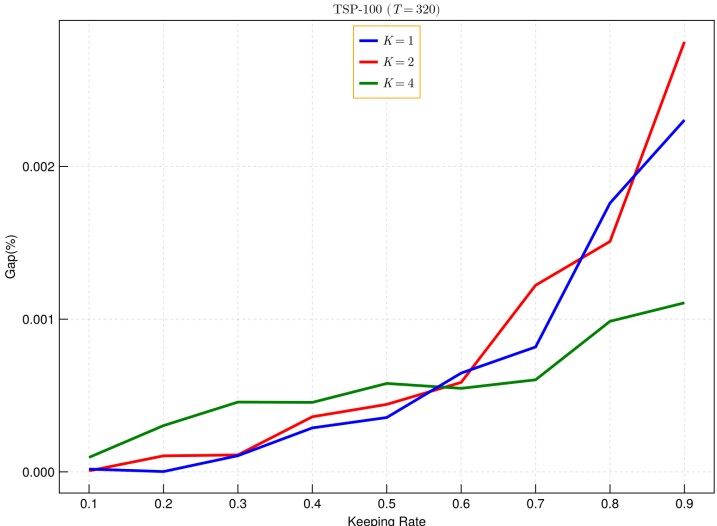

Figure 3: Grid search result of sampling steps $K$ and keeping rate $p$ on TSP-100. (T=320)

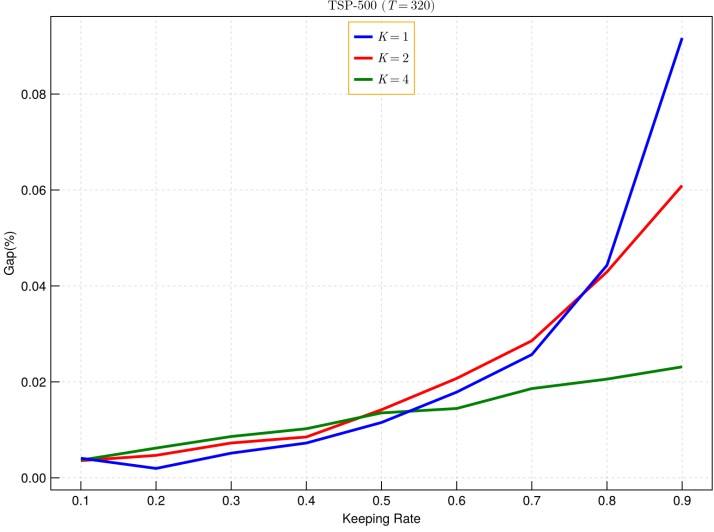

Figure 4: Grid search result of sampling steps $K$ and keeping rate $p$ on TSP-500. (T=320)

## F.4 TRAINING RESOURCE COMPARISON

The default training duration was set to 600 epochs for all models, with the exception of models trained on CVRP-500 and CVRP-1000, which were trained for 768 epochs, and those on TSP-1000, which were trained for 300 epochs. We present a direct comparison of training time on TSP with T2T and Fast T2T (Li et al., 2024) in Table 25. Notably, MaskCO requires less than $1/13$ of the training time needed by Fast T2T on TSP-1000. T2T and Fast T2T train a 12-layer GCN (for 50 epochs), whereas we train a 22-layer transformer. This comparison also reflects that the computational cost of GCN is much expensive than transformer.

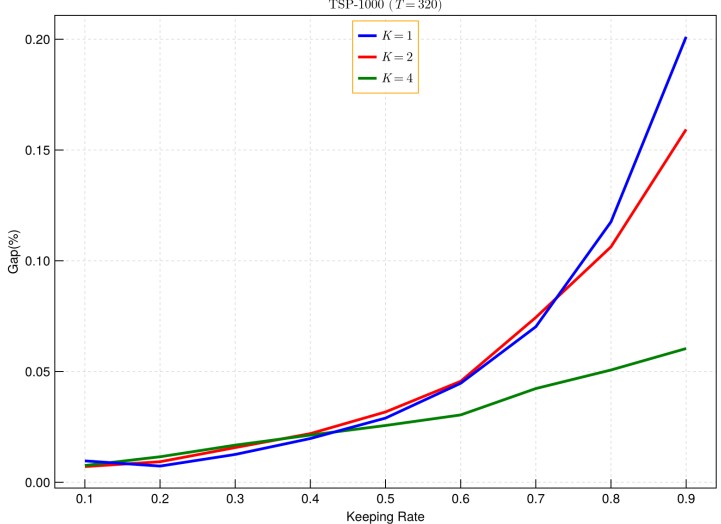

Figure 5: Grid search result of sampling steps $K$ and keeping rate $p$ on TSP-1000. (T=320)

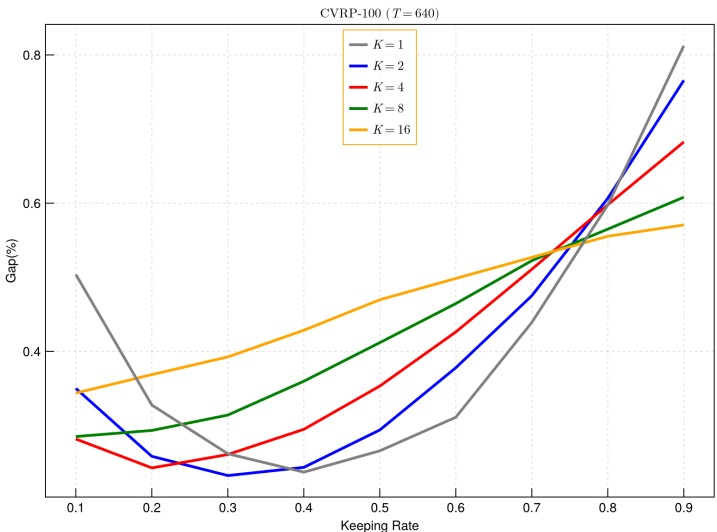

Figure 6: Grid search result of sampling steps $K$ and keeping rate $p$ on CVRP-100. (T=640)

Table 25: Comparison of training resource usage for TSP using a single A100 GPU.

|          | TSP-100    | TSP-500    | TSP-1000   |
| -------- | ---------- | ---------- | ---------- |
| T2T      | 8.6Day     | 2.7Day     | 5.1Day     |
| Fast T2T | 20.3Day    | 5.9Day     | 13.5Day    |
| MaskCO   | $\leq$2Day | $\leq$1Day | $\leq$1Day |

## F.5 TRAINING TIME FOR THE SELF-TRAINING SCHEME

For TSP-100 and TSP-500, the self-training scheme takes about $1/2$ time of the standard training time. The initial training uses the same setup as standard training but converges in just 150 epochs,

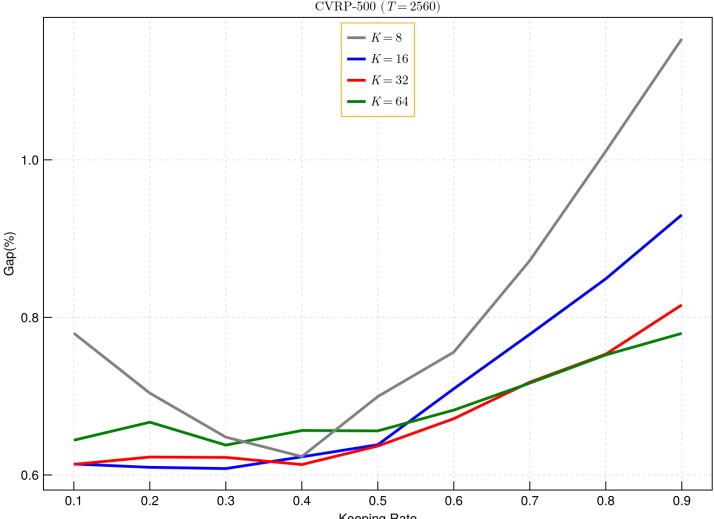

Figure 7: Grid search result of sampling steps $K$ and keeping rate $p$ on CVRP-500. (T=2560)

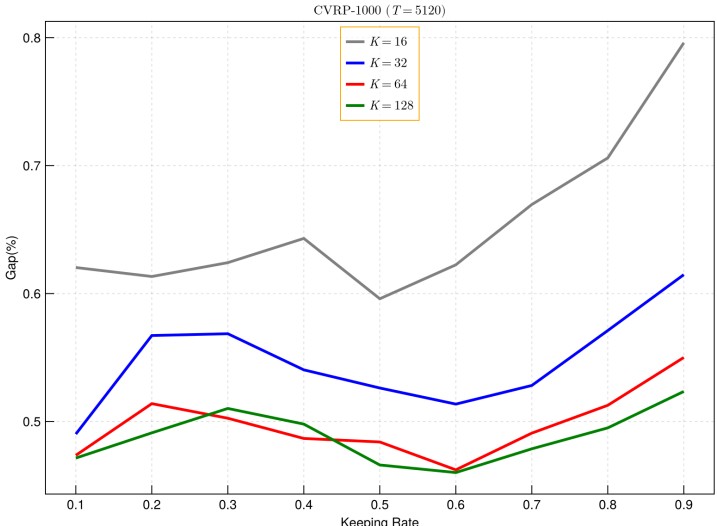

Figure 8: Grid search result of sampling steps $K$ and keeping rate $p$ on CVRP-1000. (T=5120)

only about a quarter of the 600 epochs required in the standard approach. An additional $1/4$ of the time is spent on labeling (i.e., solving unlabeled instances with the model), while each fine-tuning step takes less than 10 minutes, as overfitting occurs within 15 epochs. For TSP-1000, no initial training is needed, reducing total time to approximately $1/4$ of the standard training cost.

## F.6 EXPERIMENTS ON TSPTW

MaskCO has adaptability to more complex problems. To validate MaskCO's adaptability to more complex problems, we conduct experiments on the Traveling Salesman Problem with Time Windows (TSPTW). We conduct experiments on TSPTW-50, evaluating performance on both medium- and hard-level variants, following (Bi et al., 2024). In addition to the metrics reported in (Bi et al., 2024), we also include the constraint violation (averaged over infeasible solutions). Test sets are taken from (Bi et al., 2024). All results for MaskCO are averaged across three different seeds. Note that MaskCO adopts 2-opt with penalty factor $c = 200$. As demonstrated by Table 26 and 27, compared to SOTA neural baselines specifically designed for complex constraint handling, MaskCO achieves

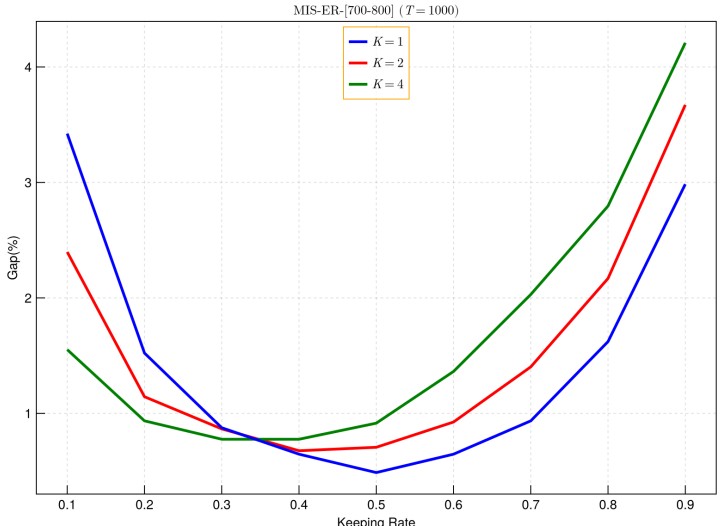

Figure 9: Grid search result of sampling steps $K$ and keeping rate $p$ on MIS-RB-[200-300]. (T=1k)

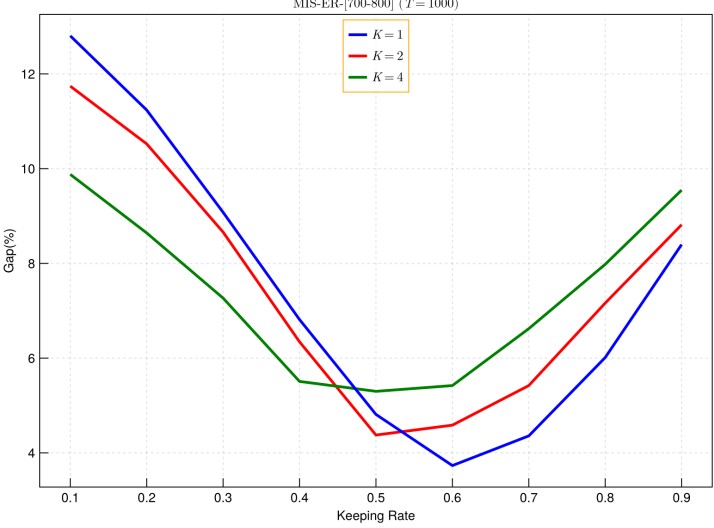

Figure 10: Grid search result of sampling steps $K$ and keeping rate $p$ on MIS-ER-[700-800]. (T=1k)

Table 26: Results on TSPTW-50 (Medium).

| METHOD | TSPTW-50 (Medium) | | | | |
|---|---|---|---|---|---|
| | INFEASIBLE INST. | OBJ. | GAP | VIOLATION | TIME |
| LKH-3* (Helsgaun, 2017) | 0.00% | 13.02 | 0.00% | – | 7h |
| 2Opt | 85.38% | 14.830 | 13.973% | 12.393 | – |
| AM+PIP* (Bi et al., 2024) | 0.35% | 13.68 | 5.06% | – | 11m |
| AM+PIP-D* (Bi et al., 2024) | 0.33% | 13.65 | 4.87% | – | 11m |
| POMO+PIP* (Bi et al., 2024) | 0.90% | 13.40 | 2.91% | – | 15s |
| POMO+PIP-D* (Bi et al., 2024) | 0.65% | 13.45 | 3.32% | – | 15s |
| MaskCO (T=320) | 0.040% | 12.918 | -0.779% | 3.24e-4 | 56s |
| MaskCO (T=640) | 0.030% | 12.872 | -1.131% | 3.26e-4 | 1m46s |
| MaskCO (T=1280) | 0.027% | 12.834 | -1.425% | 2.50e-4 | 3m26s |

up to a 10-fold reduction in the infeasibility ratio and outperforms LKH-3 in solution quality for feasible solutions, as demonstrated by the negative gap.

Table 27: Results on TSPTW-50 (Hard).

| METHOD | TSPTW-50 (Hard) | | | | |
|---|---|---|---|---|---|
| | INFEASIBLE INST. | OBJ. | GAP | VIOLATION | TIME |
| LKH-3* (Helsgaun, 2017) | 0.12% | 25.61 | 0.00% | – | 7h |
| 2Opt | 100.00% | – | – | 74.748 | – |
| AM+PIP* (Bi et al., 2024) | 1.98% | 25.71 | 0.38% | – | 11m |
| AM+PIP-D* (Bi et al., 2024) | 4.40% | 25.80 | 0.67% | – | 11m |
| POMO+PIP* (Bi et al., 2024) | 2.67% | 25.66 | 0.18% | – | 15s |
| POMO+PIP-D* (Bi et al., 2024) | 3.07% | 25.69 | 0.28% | – | 15s |
| MaskCO (T=320) | 1.353% | 25.495 | -0.459% | 0.389 | 46s |
| MaskCO (T=640) | 1.307% | 25.476 | -0.534% | 0.403 | 1m24s |
| MaskCO (T=1280) | 1.280% | 25.448 | -0.640% | 0.411 | 2m41s |

## F.7 VISUALIZATION OF MULTI-STEP DECODING

Figure 11 illustrates the multi-step decoding process of MaskCO on the TSP-500 instance. The model demonstrates an emergent ability to conduct local or global decisions across different stages. Specifically, the initial two steps are primarily dedicated to local decision-making, while the subsequent two steps transition to executing a global decision informed by the previously formulated structure.

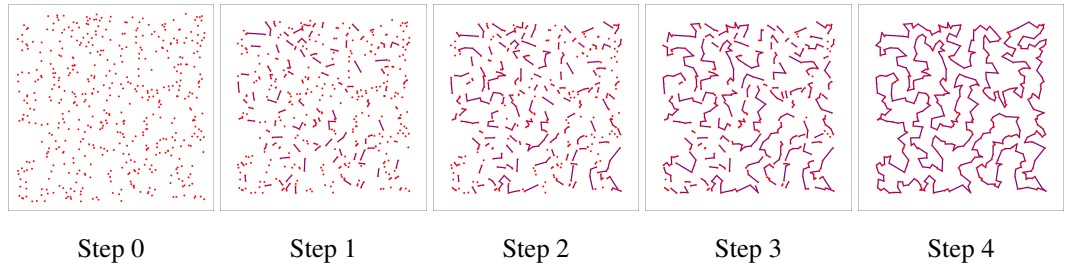

| Step 0 | Step 1 | Step 2 | Step 3 | Step 4 |

Figure 11: Visualization of multi-step decoding on TSP-500. (K=4)

