# OpenReview forum: "MaskCO: Masked Generation Drives Effective Representation Learning and Exploiting for Combinatorial Optimization"
_ICLR.cc/2026/Conference — ICLR 2026 Poster_

### Official Review · Reviewer_34zW · 2025-10-26

**Soundness:** 3
**Presentation:** 2
**Contribution:** 3
**Rating:** 6
**Confidence:** 4

**Summary:**

This paper presents MaskCO, a masked generation paradigm that formulates the learning process of neural combinatorial optimization (NCO) as a solution-level self-supervised learning framework. Specifically, it masks part of an optimal solution and reconstructs it to learn fine-grained, localized decision patterns. During inference, the model constructs a complete solution through a mask-and-reconstruct procedure, resembling a local-search-like refinement: in each iteration, certain variables are masked and regenerated, progressively improving the current solution. Extensive experiments on TSP, CVRP, and MIS demonstrate its superiority over previous baselines.

**Strengths:**

* The topic of this paper is exciting and challenging, exploring a foundational training paradigm that enables effective and scalable representation learning for CO.
* Self-supervised training is an appealing and promising paradigm for NCO.
* The reported training overhead appears lightweight, as shown in Table 23.
* The empirical results are strong.

**Weaknesses:**

* The authors claim that the proposed masked generation serves as a `foundational paradigm` for NCO. Does “foundational” here imply the goal of developing a foundation model for CO? If so, why not consider the multi-task training setting? Moreover, the paper does not discuss recent efforts toward multi-task or foundation models for CO, such as [1–4].
* The generality of the proposed approach is unclear. This work addresses TSP, CVRP, and MIS, which do not involve complex constraints. Could the proposed method handle more complex constrained VRPs [5] or other CO problems as studied in [1]?
* The training process still requires high-quality solutions as labels. Have the authors explored self-improvement learning as studied in [6]?
* The paper emphasizes representation learning. Could the authors provide a deeper analysis of the learned representations?
* The writing of this paper could be improved:
  * Parts of the introduction appear overly generated (possibly by LLMs). I would expect more direct and concrete opinions from the authors, rather than an overly abstract and ambitious presentation.
  * The descriptions in Sections 3 and 4 are somewhat verbose and make the approach seem more complicated than it is. A clear figure illustrating the overall process would improve readability.
  * It would be helpful to fully elaborate the model architecture in mathematical form.
  * Visualizing how the (partial) solution evolves through the decoding process would make the approach more intuitive.
* Minor comments:
  * Line 79: “instancesolution” → missing space.
  * Line 80: “scalabilityparticularly” → missing space.
  * Line 84: Clarify “BETR and ?”.

[1] GOAL: A Generalist Combinatorial Optimization Agent Learner. ICLR 2025.
[2] MVMoE: Multi-Task Vehicle Routing Solver with Mixture-of-Experts. ICML 2024.
[3] RouteFinder: Towards Foundation Models for Vehicle Routing Problems. TMLR 2025.
[4] UniCO: On Unified Combinatorial Optimization via Problem Reduction to Matrix-Encoded General TSP. ICLR 2025.
[5] Learning to Handle Complex Constraints for Vehicle Routing Problems. NeurIPS 2024.
[6] Boosting neural combinatorial optimization for large-scale vehicle routing problems. ICLR 2025.

----

Overall, the studied topic of this paper is exciting and challenging. I believe it makes a valuable contribution to the NCO community, and therefore I recommend acceptance.

**Questions:**

* Can the proposed method ensure 100% solution feasibility? If so, please explain how. If not, the feasibility rate should be reported in the main experimental table.
* For the TSP case in Section 4.2 (MultiStepDecoding), is the $|U(G)|=m^2$? The proposed approach seems to generate multiple dynamic heatmaps rather than a single static one as in previous methods. If so, why is the inference time of MaskCO significantly lower than that of DIFUSCO and Fast T2T? Moreover, is the decoding process conceptually similar to that used in diffusion-based LLMs?

---

> ### Author Response · Authors · 2025-11-21
> **Respond to Reviewer 34zW (Part 1/3)**
>
> Thanks for the insightful and detailed comments, and for acknowledging our work. Below we respond to your comments.
>
> ---
>
>
> > **Q1: The authors claim that the proposed masked generation serves as a foundational paradigm for NCO. Does “foundational” here imply the goal of developing a foundation model for CO? If so, why not consider the multi-task training setting? Moreover, the paper does not discuss recent efforts toward multi-task or foundation models for CO, such as [1–4].**
>
>
>
> Thanks for your nice questions. **"Foundational paradigm" refers to the core generation approach (e.g., autoregressive (AR), diffusion, or masked generation), NOT the concept of a foundation model.** These paradigms operate at a lower level: they define how solutions are constructed during inference. In contrast, foundation models represent a higher-level architectural and training philosophy (e.g., multi-task pretraining, transfer learning), which is largely orthogonal to the choice of generation paradigm. For instance, one could replace the autoregressive generation in a framework like GOAL or MVMoE with our masked generation paradigm to obtain a variant such as GOAL-Mask or MVMoE-Mask, potentially benefiting from the improved solution construction capabilities of masking.
>
> Also, thank you for noting recent efforts toward multi-task or foundation models for CO, we add this part in the Related Works part of the updated manuscript. We believe that our improved generation paradigm can significantly facilitate future studies aimed at developing effective foundation models for CO problems by offering a superior mechanism for solution generation.
>
>
> > **Q2: The generality of the proposed approach is unclear. This work addresses TSP, CVRP, and MIS, which do not involve complex constraints. Could the proposed method handle more complex constrained VRPs [5] or other CO problems as studied in [1]?**
>
>
>
>
>
>
> Thanks for the insightful review. A key limitation of heatmap-based methods lies in their inherent difficulty in modeling complex constraints, such as the capacity constraint of CVRP. This is why these methods continues to refresh SOTAs on TSP and MIS, but is still limited in the scope of classical graph problems. Our MaskCO has successfully addressed this fundamental limitation, as evidenced by its effective handling of the CVRP. Moreover, MaskCO is ready to extend to even more complex problems like TSPTW:
>
>
> We conduct experiments on TSPTW-50, evaluating performance on both medium- and hard-level variants, following [5]. In addition to the metrics reported in [5], we also include the constraint violation (averaged over infeasible solutions). Test sets are taken from [5]. All results for MaskCO are averaged across three different seeds.
>
> **TSPTW-50 medium**
> || Infeasible Inst. | Obj.| Gap.      | Violation | Time  |
> |-| -|-| --------- |-| - |
> | LKH3| 0.00%| 13.02      | 0.00%| --| 7h    |
> | 2Opt| 85.38%| 14.830     | 13.973 %  | 12.393    | --    |
> | AM+PIP| 0.35%| 13.68| 5.06%| --| 11m   |
> | AM+PIP-D| 0.33%| 13.65| 4.87%| --| 11m   |
> | POMO+PIP| 0.90 %| 13.40| 2.91%| --| 15s   |
> | POMO+PIP-D      | 0.65 %| 13.45| 3.32%     | --| 15s   |
> | MaskCO (T=320)  | 0.040 %| 12.918| -0.779%   | 3.24e-4   | 56s   |
> | MaskCO (T=640)  | 0.030 %| 12.872     | -1.131%   | 3.26e-4   | 1m46s |
> | MaskCO (T=1280) | **0.027 %**| **12.834** | **-1.425%** | 2.50e-4   | 3m26s |
>
>
> **TSPTW-50 hard**
> |                 | Infeasible Inst. | Obj.       | Gap.        | Violation | Time  |
> | -| - | ---------- | ----------- | --------- | ----- |
> | LKH3| 0.12%| 25.61      | 0.00%       | --        | 7h    |
> | 2Opt            | 100.00%| --| --          | 74.748    | --    |
> | AM+PIP          | 1.98%| 25.71      | 0.38%       | --        | 11m   |
> | AM+PIP-D        | 4.40%| 25.80      | 0.67%       | --        | 11m   |
> | POMO+PIP        | 2.67%| 25.66      | 0.18%       | --        | 15s   |
> | POMO+PIP-D      | 3.07%| 25.69      | 0.28%       | --        | 15s   |
> | MaskCO (T=320)  | 1.353%| 25.495     | -0.459%     | 0.389     | 46s   |
> | MaskCO (T=640)  | 1.307%| 25.476     | -0.534%     | 0.403     | 1m24s |
> | MaskCO (T=1280) | **1.280%**| **25.448** | **-0.640%** | 0.411     | 2m41s |
>
> Note that MaskCO adopts 2-opt with penalty factor $c=200$. **Compared to SOTA neural baselines specifically designed for complex constraint handling, MaskCO achieves up to a 10-fold reduction in the infeasibility ratio and outperforms LKH-3 in solution quality for feasible solutions, as demonstrated by the negative gap.** For TSPTW, 2-opt alone is not an effective end-to-end constraint handler, as evidenced by its high infeasibility ratio. Instead, it functions as a "projection" operator, transforming solutions produced by masked generation from the relaxed space into feasible solutions.
>
> We are also extending MaskCO to the Orienteering Problem (OP) as studied in [1]. Results will be added once the experiments are completed.

---

> ### Author Response · Authors · 2025-11-21
> **Respond to Reviewer 34zW (Part 2/3)**
>
> > **Q3: The training process still requires high-quality solutions as labels. Have the authors explored self-improvement learning as studied in [6]?**
>
>
>
> Yes, in Section 5.4 of the original manuscript, self-improvement learning is studied: even with simple self-training strategy, MaskCO reaches 0.000% on TSP-100, and surpasses Fast T2T on TSP across all scales.
>
> This self-improvement is currently achieved through a two-stage iterative strategy:
>
> * **Initialization/Warmup:** The process begins by using low-quality heuristic labels (specifically, 2-opt with 128 restarts) to initialize the model.
> * **Self-Improvement Iteration:** The model's ability to **outperform its initial imitation source** is a pivotal moment in the self-improvement process. From that point, the model enters a continuous self-training loop: it alternates between generating high-quality pseudo-labels for previously unlabeled problem instances and performing lightweight fine-tuning on these newly generated labels.
>
> It can be further enhanced by formulating it as a reinforcement learning problem or by adopting more advanced self-training techniques like [6]. For your convenience in locating it, we have highlighted this section of the experiment and its description in blue.
>
>
> > **Q4: The paper emphasizes representation learning. Could the authors provide a deeper analysis of the learned representations?**
>
>
>
> Thank you for the insightful question. The focus on representation learning is indeed central to our work, as it underpins the model's adaptability and generalization. Our MaskCO framework excels at learning powerful, general representations. Beyond its primary masked decoding mechanism, MaskCO automatically develops a strong feature set for a given combinatorial problem instance. This crucial capability allows the learned representations to be immediately effective when used with various inference techniques, such as AR, Diffusion, Consistency, or Monte Carlo Tree Search (MCTS).
>
>
> We acknowledge a current limitation: direct visualization or quantitative analysis of the learned representations is challenging. Since CO problem instances lack a meaningful classification structure, projecting them into a low-dimensional space for visual inspection is not feasible. We defer addressing these representation analysis techniques to future work. Nevertheless, strong indirect evidence supports the quality and generality of the learned representations: they enable seamless adaptation to diverse inference strategies and facilitate rapid generalization to novel ones:
>
> * **Consistency Formulation Transfer.** As shown in Table 6, using only one epoch of few-shot fine-tuning on the representation significantly improves performance under the Consistency formulation. This formulation conditions on discretely noised solutions, a pattern the model did not encounter during its original training (which used partial solutions). This remarkable transferability to an entirely new algorithmic setup confirms the representations are general and reusable.
>
> * **Handling Novel Input Patterns.** When applying the model to AR decoding for TSP, it is required to reason about a contiguous tour segment. During the original training with random masking, the model rarely encountered this specific input pattern, as random masking typically produces many disconnected segments. The model's ability to successfully handle and complete these contiguous segments via AR decoding further proves that its representations capture the fundamental structure of the problem, not just the specific masking scheme used during pre-training.

---

> ### Author Response · Authors · 2025-11-21
> **Respond to Reviewer 34zW (Part 3/3)**
>
> > **Q5: Can the proposed method ensure 100% solution feasibility? If so, please explain how. If not, the feasibility rate should be reported in the main experimental table.**
>
> Yes, MaskCO can ensure 100% solution feasiblility on problems included in the original manuscript (i.e., TSP, MIS, CVRP).
>
> For TSP and MIS, since we do not use relaxation on solution space, the feasibility can be directly guaranteed by the selection function, or more concretely, the "greedy insertion" used in DIFUSCO and Fast T2T.
>
> For CVRP, we adopt a capacity-relaxed solution space by default. Consequently, the selection function generates solutions that satisfy all constraints except capacity. To enforce capacity feasibility, we apply an additional 2-opt procedure with penalty terms, which guarantees that all CVRP solutions satisfy the capacity constraints. Unlike TSPTW, the capacity constraint in CVRP can always be enforced through 2-opt. The 2-opt procedure for CVRP is detailed below:
>
> * **Penalty terms are directly incorporated into the objective function**:
> $$
> l_{\mathrm{penalized}}(\mathbf{x};G)=l(\mathbf{x};G)+c\cdot \mathrm{violation}(\mathbf{x};G),
> $$
>
> * **A sufficient number of empty subroutes are appended.**
>
> At each iteration of the 2-opt procedure, the algorithm evaluates all feasible 2-opt moves and selects the one that yields the greatest reduction in $l_{\mathrm{penalized}}$. The selected move is applied only if it results in a negative change, i.e., $\Delta l_{\mathrm{penalized}} < 0$. For CVRP, by setting $c$ sufficiently large, the algorithm is incentivized to eliminate all capacity violations, as any subroute violating capacity constraints can be repaired by splitting it into two or more subroutes (by applying a 2-opt move on it and an empty subroute).
>
>
> > **Q6: For the TSP case in Section 4.2 (MultiStepDecoding), is the $|U(G)|=m^2$? The proposed approach seems to generate multiple dynamic heatmaps rather than a single static one as in previous methods. If so, why is the inference time of MaskCO significantly lower than that of DIFUSCO and Fast T2T? Moreover, is the decoding process conceptually similar to that used in diffusion-based LLMs?**
>
> Thank you for your thoughtful questions. For the TSP case, $|U(G)|=m(m-1)/2=O(m^2)$. The improvement on single-pass evaluation time is basically from the architecture, where the GCN used in DIFUSCO and Fast T2T is replaced by transformer. Ablation studies on architecture (Table 7) illustrates that using Transformer leads to much faster inference for the same decoding process.
>
>
> Conceptually, the decoding process of MaskCO is indeed similar to that used in diffusion LLMs. Both approaches are inspired by MaskGIT [7], a method originally developed for image generation that uses masked modeling for parallel iterative generation. This powerful generative framework is then adapted to structured prediction tasks such as text generation and combinatorial optimization problems. While sharing the same core idea, these methods differ in their task-specific adaptations.
>
>
> ---
>
> We hope this response could help address your concerns, and wish to receive your further feedback soon. We are more than happy to address any further concerns you may have.
>
>
>
> ------
>
> [1] GOAL: A Generalist Combinatorial Optimization Agent Learner. ICLR 2025.
>
> [2] MVMoE: Multi-Task Vehicle Routing Solver with Mixture-of-Experts. ICML 2024.
>
> [3] RouteFinder: Towards Foundation Models for Vehicle Routing Problems. TMLR 2025.
>
> [4] UniCO: On Unified Combinatorial Optimization via Problem Reduction to Matrix-Encoded General TSP. ICLR 2025.
>
> [5] Learning to Handle Complex Constraints for Vehicle Routing Problems. NeurIPS 2024.
>
> [6] Boosting neural combinatorial optimization for large-scale vehicle routing problems. ICLR 2025.
>
> [7] MaskGIT: Masked Generative Image Transformer. CVPR 2022.

---

> > ### Comment · Reviewer_34zW · 2025-11-24
> >
> > Thanks for your rebuttal, which addresses most of my concerns. I have three remaining comments:
> > * Since ICLR allows paper modification, I would like to see how the authors address my comments regarding the paper writing. In addition, the newly added experiments do not appear in the revised version. Please include them as well.
> > * For Q1 and Q2: Further discussion on the applicability of the proposed paradigm would be valuable. For instance, it would be helpful to clarify which types of CO problems/domains (e.g., routing, scheduling, etc.) the approach finds challenging. What are the technical challenges in training a multi-task model under the proposed paradigm? Such insights would be valuable for guiding future work on adopting this paradigm for multi-task or even foundation model development.
> > * Will the authors release the source code?

---

> > > ### Author Response · Authors · 2025-11-28
> > > **Response to Reviewer 34zW**
> > >
> > > Sincerely, thank you for your follow-up and your valuable suggestions you provided for enhancing our work! Your suggestions are crucial for the refinement and improvement of our work. We provide the following clarifications in response to your comments abd hope this will address your concerns.
> > >
> > > --------------------
> > >
> > > > Suggections regarding the paper writing.
> > >
> > > Thanks for your nice suggestions.
> > >
> > > * For the mathematical formulation of the model, we add it in Appendix Q.
> > >
> > > * For visualizing how the partial solution evolves through the decoding process, we add it in Appendix R.
> > >
> > > * For a clear figure illustrating the overall process, we add Figure 2 which illustrates the inference process, complemented by Figure 1 which is for training process.
> > >
> > > * For more direct and concrete opinions in the introduction part, we sincerely apologize for the potentially ambiguous statement. We have modified the introduction, particularly on the single/multi-task clarification and more detailed inference description with additional figures, which are mentioned by the reviewer.
> > >
> > >
> > >
> > > > Include the newly added experiments in the manuscripts.
> > >
> > > We have added these experiments in Appendix G, N, and P.
> > >
> > > > Further discussion on the applicability of the proposed paradigm with regard to the types of CO problems. Technical challenges in training a multi-task model under the proposed paradigm.
> > >
> > > Models trained with masked generation paradigm has the ability to conduct generation with AR decoding, as evidenced by Table 6. As a result, in theory, MaskCO covers the CO problems which can be handled by AR modeling, via disabling global parallel decoding if a selection function is hard to implemented. Note that AR decoding with MaskCO can be also parallelized by applying AR-style mask on the heatmap once a new node is selected, which is not possible in previous AR works as they only produce probabilites related to the current node.
> > >
> > > However, operating in this AR decoding mode holds at the cost of possible performance degradation. For adapting a full-version MaskCO to a new type of CO problem, the primary challenge lies in implementing the selection function. An easier implementation can be achieved through appropriate relaxation, often inspired by the relaxation methods used in traditional search algorithms for that specific problem type.
> > >
> > > **Vehicle Routing Problems (VRPs).** We recommand relaxing all VRP-specific contraints and reducing the problem to a Traveling Salesperson Problem (TSP) or multiple Traveling Salesman Problem (mTSP) constraint, or alternatively, a Hamiltonian cycle constraint. LKH-3 serves as an good case study; it is built upon the TSP-specific LKH-2 and incorporates task-specific penalty terms to re-introduce the original VRP constraints.
> > >
> > > **Scheduling Problems.** While we have not tried this type of problems, some possible relaxation strategies are presented here. For instance, for the Job Shop Scheduling Problem (JSSP), we can model operations as nodes, and an edge between two nodes exists if and only if they are consective on the same machine; the constraint of ensuring no cyclic dependency is offloaded to a local search.
> > >
> > > **Graph-Based Problems.** The constraints for these problems are relatively easy, so no relaxation is necessary.
> > >
> > > Another practical issue beyond theoretical adaptability is how *well* MaskCO performs across different problem types. While experiments on VRPs and graph-based tasks have yielded promising results, its performance on scheduling problems and other CO domains remains underexplored. This represents a current limitation, but we will continue to extend MaskCO to more domains.
> > >
> > > For the technical challenges found in training a multi-task model under masked generation paradigm, we think the primary and the most important requirement is that the paradigm can effectively handle any one of the tasks. Again, at the training stage, compared with AR, masked modeling applies uniform mask instead of contiguous mask, uses global prediction as training target instead of local prediction, while others are same, so we think many properties of MaskCO on multi-task learning are similar to AR. For architecture, some enhancement specified for multi-task learning in GOAL and MVMoE can be inherited, since their proposed transformer-based architectures are compatible with additive attention bias.
> > >
> > >
> > > > Will the authors release the source code?
> > >
> > > Yes, we will release the source code once accepted.
> > >
> > > ---------------
> > >
> > > Let us know whether you have further questions or suggestions. Please do not hesitate to ask, we would be more than happy to further clarify our presentation to strengthen our paper.

---

### Official Review · Reviewer_vnPK · 2025-10-31

**Soundness:** 2
**Presentation:** 3
**Contribution:** 3
**Rating:** 4
**Confidence:** 3

**Summary:**

The paper proposes MaskCO, a method that masks parts of optimal solutions and trains policies to reconstruct them. Through experiments on the TSP, CVRP, and MIS problems, the authors show the efficacy of their approach.

**Strengths:**

- The paper is well-written and easy to read.
- The proposed approach, MaskCO, tackles neural CO with a novel approach: mask portions of the solutions to generate more data.

**Weaknesses:**

- My primary concern with the paper is that it requires expert solutions. In general, the community is moving away from supervised learning based methods towards RL.
- The paper does not consider several SOTA baselines [1, 2].

[1] Grinsztajn et al. Winner Takes It All: Training Performant RL Populations for Combinatorial Optimization, NeurIPS 2023.

[2] Hottung et al. PolyNet: Learning Diverse Solution Strategies for Neural Combinatorial Optimization, ICLR 2025.

**Questions:**

See weaknesses.

---

> ### Author Response · Authors · 2025-11-21
> **Response to Reviewer vnPK (Part 1/2)**
>
> Thanks for appreciating the novelty of our work. We value your insightful concerns and have made every effort to address them thoroughly. Our detailed responses are provided below.
>
>
> ---
>
> > **Q1: My primary concern with the paper is that it requires expert solutions. In general, the community is moving away from supervised learning based methods towards RL.**
>
>
> Thank you for your question.
>
> In Section 5.4, we investigate MaskCO’s compatibility with a self-training paradigm---an RL-like setting that does not rely on expert solutions and allows the model to iteratively improve itself. The results are striking: with our current (admittedly naive) self-training scheme, MaskCO achieves a **0.000% gap on TSP-100** and **achieves SOTA without expert solutions on TSP**.
>
> This self-training is currently achieved through a two-stage iterative strategy:
>
> * **Initialization/Warmup:** The process begins by using low-quality heuristic labels (specifically, 2-opt with 128 restarts) to initialize the model.
> * **Self-Improvement Iteration:** The model's ability to **outperform its initial imitation source** is a pivotal moment in the self-improvement process. From that point, the model enters a continuous self-training loop: it alternates between generating high-quality pseudo-labels for previously unlabeled problem instances and performing lightweight fine-tuning on these newly generated labels.
>
> The process begins with Behavior Cloning (Stage 1), which is then followed by a second stage that leverages concepts from offline RL for policy refinement. We believe this framework can be substantially enhanced by formulating self-training more rigorously as an RL problem (like Self-Imitation Learning [3] and ReST [4]) or by integrating more advanced self-improvement techniques, which is another compelling avenue for future research.
>
> Moreover, we’d like to clarify that **both RL and SL collaboratively push the boundaries of learning-based solvers**. **Pure RL** offers the advantage of **not relying on high-quality labels**, making it more easily **adaptable** to new problems. Conversely, **pure SL** excels by **exploiting existing in-domain experts and knowledge** on classical problems, often leading to superior performance on well-studied problems.
>
> **We view SL and RL as complementary rather than competing approaches.** A combination of RL and SL is promising but remains understudied. Two key approaches include:
>
> 1.  **Pre-training using SL** and **fine-tuning** on new settings using **RL**.
> 2.  Training an **RL-based model as an expert** on new problems, and using **SL-based methods to distill this knowledge** by training on the solutions generated by the RL expert.
>
> The pre-training and post-training paradigm of Large Language Models (LLMs) serves as a successful example of this combination. The full potential of RL/SL synergy in CO can only be realized once both techniques are thoroughly understood and emphasized.

---

> ### Author Response · Authors · 2025-11-21
> **Response to Reviewer vnPK (Part 2/2)**
>
> > **Q2: The paper does not consider several SOTA baselines [1, 2].**
>
> Thank you for bringing this to our attention. We have updated the latest manuscript to include these baselines: For [2]: We evaluated this model using its publicly available published checkpoints; For [1]: Since a publicly available checkpoint could not be located, we have quoted the results directly from the [1] paper.
>
> For your convenience, we present the results of these baselines here below:
>
> **TSP-100**
> | Method| Obj.| Gap| Time|
> |-|-|-|-|
> | Poppy (16)| 7.770| 0.07%| 1m|
> | PolyNet (sampling) | 7.756| 0.0006%| 0.36m|
> | PolyNet (EAS)| **7.756** | **0.0000%** | 20.66m |
> | MaskCO (T=320)| **7.756** | **0.0000%** | 8s|
>
>
> **CVRP-100**
> | Method| Obj.| Gap| Time|
> |-|-|-|-|
> | Poppy (32)| 15.73| 1.06%| 5m|
> | PolyNet (sampling)| 15.627| 0.496%|0.46m|
> | PolyNet (EAS)| 15.571|0.139%| 25.40m |
> | MaskCO (T=640)| 15.586|0.232%| 32s|
> | MaskCO (T=2560)| 15.571|0.135%|2m5s|
> | MaskCO (T=10240)|**15.563**| **0.086%** | 8m17s  |
>
> MaskCO still demonstrates clear advantages over these SOTA baselines, including the recent PolyNet presented at ICLR 2025. For the TSP-100 problem, MaskCO achieves a 0.000% Gap with an inspected speedup of approximately 150x compared to PolyNet (EAS). On CVRP-100, MaskCO (T=640) achieves a gap of 0.232%, which is less than half the gap of PolyNet (sampling) (0.496%), while operating within a comparable inference time (32s vs. 27.6s). When given a higher computational budget (T=2560), MaskCO matches the solution quality of PolyNet (EAS), while remaining over 12x faster.
>
> ---
>
> We hope this response could help address your concerns. We believe that this work contributes to this community with its strong empirical performance and seamless compatibility with both SL and RL-like training paradigms. We would sincerely appreciate it if you could reconsider your rating and we are more than happy to address any further concerns you may have.
>
>
>
> -----
>
>
> [1] Grinsztajn et al. Winner Takes It All: Training Performant RL Populations for Combinatorial Optimization, NeurIPS 2023.
>
> [2] Hottung et al. PolyNet: Learning Diverse Solution Strategies for Neural Combinatorial Optimization, ICLR 2025.
>
> [3] Oh et al. Self-Imitation Learning, ICML 2018.
>
> [4] Gulcehre et al. Reinforced Self-Training (ReST) for Language Modeling.

---

### Official Review · Reviewer_VktH · 2025-10-31

**Soundness:** 3
**Presentation:** 2
**Contribution:** 3
**Rating:** 4
**Confidence:** 5

**Summary:**

This paper introduces MaskCO, a new method that uses a masked generation approach, similar to techniques in NLP and computer vision. Instead of generating a whole solution at once, the model is trained to fill in missing parts of known good solutions, which helps it learn the important local patterns within those solutions. The experiments show that MaskCO works well on three different COP tasks:  the Traveling Salesman Problem (TSP), Capacitated Vehicle Routing Problem (CVRP), and Maximum Independent Set (MIS). During inference, it employs the 2-opt local search to further enhance the performance.

**Strengths:**

1. The self-supervised paradigm of masked generation in CO is interesting, although it was first noted in [1].
2. The performance, easpecially on large-scale CVRP, is impressive.


[1] Solving Diverse Combinatorial Optimization Problems with a Unified Model.

**Weaknesses:**

1. The idea of masked generation for CO is similar to that in [1].

2. Some equations and definitions in Sections 3 and 4 could be simplified; currently, they are unnecessarily complicated, which reduces the paper’s readability.

3. The implementation details for the 2-opt heuristic are unclear. How does it, with the use of penalty terms, enforce constraint satisfaction?

4. The effect of adding 2-opt appears minor for TSP but significantly different for CVRP, as shown in Table 11. Most importantly, on CVRP-500 and CVRP-1000, without 2-opt, MaskCO fails to generate feasible solutions. This raises concerns about its adaptability to more complex problems.

5. The hyperparameters (e.g., $K$ and $p$) vary across problem types and sizes, as shown in Tables 12–15. It would be helpful to provide the rationale behind these choices. The results in Figures 2–9 demonstrate that model performance is highly sensitive to these hyperparameters.

**Questions:**

How is MaskCO scalable? Any designs and empirical results to support this claim?

---

> ### Author Response · Authors · 2025-11-21
> **Response to Reviewer VktH (Part 1/4)**
>
> Thank you for your valuable comments and for acknowledging the strong performance of MaskCO. We appreciate your insightful concerns and have made every effort to address them thoroughly. Our detailed responses are provided below.
>
> ---
>
> > **Q1: The idea of masked generation for CO is similar to that in [1].**
>
> Thank you for pointing out reference [1]. We agree that [1] also leverages self-supervised learning, but we respectfully clarify that its approach is fundamentally different from the *masked generation* paradigm introduced in MaskCO. Specifically, [1] adopts a *next-token prediction*-style strategy framed as a sequential decision-making process for generating complete solutions. Below, we elaborate on the key distinctions across three dimensions:
>
> 1) **Algorithmic Objective**:
>    [1] aims to enhance *cross-problem generalization* by training a single unified model to solve multiple combinatorial optimization (CO) problems simultaneously. This motivates architectural choices such as CO-prefix conditioning to enable task-aware representations and cross-task transfer. In contrast, MaskCO focuses on *pushing the limits of performance on a single problem*. It learns a powerful problem-specific representation together with a flexible decoding backbone that supports diverse inference strategies, including autoregressive sampling, consistency decoding, and iterative mask regeneration, thereby fully exploiting the learned representation. This design enables unprecedented results: on TSP-1000, MaskCO achieves a solution within 0.01% of optimality in just 18 seconds, a dramatic improvement over prior state-of-the-art methods, which required 18 minutes to reach a 0.42% gap.
>
> 2) **Self-Supervised Learning Design**:
>    MaskCO’s training objective involves *non-sequential, random masking* of arbitrary subsets of decision variables (with varying mask ratios and spatial distributions). This encourages the model to develop highly versatile decoders capable of reconstructing solutions from partial observations, a capability that seamlessly supports multiple decoding paradigms. By contrast, [1] formulates solution construction as a Markov Decision Process (MDP) and trains the model via *step-by-step next-action prediction*, effectively learning one step of a sequential generation pipeline. While both are self-supervised, they align with distinct paradigms seen in CV and NLP fields: MaskCO resembles masked autoencoders (e.g., MAE in vision or BERT in NLP), whereas [1] follows the autoregressive next-token prediction framework (e.g., GPT-style models). These represent fundamentally different inductive biases and training dynamics.
>
> 3) **Empirical Performance and Scope**:
>    The unified model in [1] makes an important contribution by demonstrating cross-problem applicability, which is a valuable step toward generalist CO solvers. However, it does not aim to surpass specialized single-problem solvers in terms of solution quality or speed. MaskCO, while currently focused on single-problem mastery rather than multi-task generality, achieves breakthrough performance on large-scale TSP, setting a new standard for both speed and accuracy. Again, on TSP-1000, MaskCO reaches sub-0.01% gaps in 18 seconds, far outpacing previous methods.
>
> We view both works as pioneering efforts in applying self-supervised representation learning to combinatorial optimization. Though their goals and technical approaches differ, they collectively advance the vision of building foundational, reusable representations for CO. We have expanded our Related Work section to better contextualize these complementary directions and acknowledge the role of [1] in this emerging landscape.
>
> > **Q2: The implementation details for the 2-opt heuristic are unclear. How does it, with the use of penalty terms, enforce constraint satisfaction?**
>
> Thank you for your attention to the inference details. The implementation of the 2-opt heuristic for CVRP can be clarified as follows: constraint satisfaction is enforced through **penalty terms incorporated directly into the objective function**:
> $$l_{\mathrm{penalized}}(\mathbf{x};G)=l(\mathbf{x};G)+c\cdot\mathrm{violation}(\mathbf{x};G),$$
> where $c$ is a penalization factor ($c=3$ for CVRP in our experiments) and $\mathrm{violation}(\mathbf{x};G)$ denotes the total capacity violation across all subroutes. **Moreover, a sufficient number of empty subroutes are appended.** At each iteration of the 2-opt procedure, the algorithm evaluates all feasible 2-opt moves and selects the one that yields the greatest reduction in $l_{\mathrm{penalized}}$. The selected move is applied only if it results in a negative change, i.e., $\Delta l_{\mathrm{penalized}} < 0$. For CVRP, by setting $c$ sufficiently large, the algorithm is incentivized to eliminate all capacity violations, as any subroute violating capacity constraints can be repaired by splitting it into two or more subroutes (by applying a 2-opt move on it and an empty subroute).

---

> ### Author Response · Authors · 2025-11-21
> **Response to Reviewer VktH (Part 2/4)**
>
> > **Q3: The effect of adding 2-opt appears minor for TSP but significantly different for CVRP, as shown in Table 11. Most importantly, on CVRP-500 and CVRP-1000, without 2-opt, MaskCO fails to generate feasible solutions. This raises concerns about its adaptability to more complex problems.**
>
>
> We sincerely appreciate the reviewer’s insightful observation.
>
> The infeasibility issues noted by the reviewer arise from our use of a capacity-relaxed solution space for CVRP, which is an intentional design choice that prioritizes solution quality during the construction phase. **Importantly, we retain the flexibility to directly employ a strictly feasible solution space when the experimental setup demand it.** We include performance results of MaskCO when constrained to a strictly feasible solution space.
>
> | w/o relaxation, w/o 2-opt | CVRP-100| CVRP-500| CVRP-1000|
> |-|-|-|-|
> | T=640| 15.661, 0.717 % | 64.60, 3.944 % | 126.36, 4.370 % |
> | T=1280| 15.630, 0.515 % | 64.22, 3.327 % | 125.61, 3.752 % |
> | T=2560| 15.608, 0.374 % | 63.87, 2.768 % | 124.82, 3.095 % |
> | T=5120| 15.593, 0.276 % | 63.60, 2.326 % | 124.17, 2.559 % |
> | T=10240| 15.583, 0.212 % | 63.32, 1.885 % | 123.91, 2.347 % |
>
>
> **If the relaxed space is not used, the constraint can be directly enforced via the selection function, eliminating the dependency on 2-opt.**
>
>
>
> We also provide the comparative studies demonstrating the performance differences in CVRP solving with and without the capacity relaxation strategy to support this design choice of capacity-relaxed space.
>
> | w/o relaxation, w/ 2-opt | CVRP-100        | CVRP-500       | CVRP-1000       |
> | ------------------------ | --------------- | -------------- | --------------- |
> | T=640| 15.598, 0.308 % | 62.91, 1.226 % | 122.65, 1.306 % |
> | T=1280| 15.587, 0.238 % | 62.81, 1.069 % | 122.36, 1.065 % |
> | T=2560| 15.578, 0.183 % | 62.75, 0.959 % | 122.16, 0.898 % |
> | T=5120| 15.572, 0.145 % | 62.69, 0.864 % | 121.97, 0.746 % |
> | T=10240| 15.568, 0.116 % | 62.63, 0.770 % | 121.95, 0.726 % |
>
>
>
> | w/ relaxation, w/ 2-opt | CVRP-100        | CVRP-500       | CVRP-1000       |
> | - | -| -| --------------- |
> | T=640| 15.586, 0.232 % | 62.66, 0.813 % | 122.03, 0.798 % |
> | T=1280| 15.577, 0.176 % | 62.59, 0.714 % | 121.85, 0.644 % |
> | T=2560| 15.571, 0.135 % | 62.53, 0.608 % | 121.69, 0.514 % |
> | T=5120| 15.567, 0.111 % | 62.47, 0.514 % | 121.63, 0.460 % |
> | T=10240| 15.563, 0.086 % | 62.43, 0.448 % | 121.60, 0.438 % |
>
> **Results shows that relaxation brings consistent improvement.** Actually, advanced traditional solvers such as HGS and LKH-3 operate within a relaxed search space, and NeuOpt has demonstrated that exploring infeasible solutions can yield benefits for learning-based search methods. We believe that introducing 2-opt at test time is a relative simple way to use the relaxed space, or some mechenisms like those used in NeuOpt should be included when training.
>
>
> **MaskCO has adaptability to more complex problems. To validate it, we conduct experiments on TSPTW.** We conduct experiments on TSPTW-50, evaluating performance on both medium- and hard-level variants, following [2]. In addition to the metrics reported in [2], we also include the constraint violation (averaged over infeasible solutions). Test sets are taken from [2]. All results for MaskCO are averaged across three different seeds.
>
>
>
> **TSPTW-50 medium**
> ||Infeasible Inst.|Obj.| Gap.| Violation | Time  |
> |-|-|-|-|-|-|
> | LKH3| 0.00%|13.02|0.00%|--|7h|
> | 2Opt| 85.38%|14.830 | 13.973% | 12.393| --    |
> | AM+PIP| 0.35%|13.68  | 5.06%|--| 11m|
> | AM+PIP-D| 0.33%|13.65  | 4.87%|--| 11m|
> | POMO+PIP| 0.90%|13.40  | 2.91%|--| 15s|
> | POMO+PIP-D| 0.65%|13.45  | 3.32%    |--| 15s   |
> | MaskCO (T=320)|0.040%| 12.918 | -0.779%  | 3.24e-4| 56s   |
> | MaskCO (T=640)|0.030%| 12.872 | -1.131%  | 3.26e-4| 1m46s |
> | MaskCO (T=1280|0.027%| 12.834 | -1.425%  |2.50e-4| 3m26s |
>
>
> **TSPTW-50 hard**
> || Infeasible Inst.|Obj.|Gap.|Violation|Time|
> |-|-|-|-|-|-|
> |LKH3|0.12%|25.61|0.00%|--|7h|
> |2Opt|100.00%|--|--|74.748|--|
> |AM+PIP|1.98%|25.71|0.38%|--|11m|
> |AM+PIP-D|4.40%|25.80|0.67%|--|11m|
> |POMO+PIP|2.67%|25.66|0.18%|--|15s|
> |POMO+PIP-D|3.07%|25.69|0.28%|--|15s|
> |MaskCO (T=320)|1.353%|25.495|-0.459%|0.389|46s|
> |MaskCO (T=640)|1.307%|25.476|-0.534% |0.403|1m24s|
> |MaskCO (T=1280)|1.280%|25.448|-0.640%|0.411|2m41s|
>
> Note that MaskCO adopts 2-opt with penalty factor $c=200$. **Compared to SOTA neural baselines specifically designed for complex constraint handling, MaskCO achieves up to a 10-fold reduction in the infeasibility ratio and outperforms LKH-3 in solution quality for feasible solutions, as demonstrated by the negative gap.** For TSPTW, 2-opt alone is not an effective end-to-end constraint handler, as evidenced by its high infeasibility ratio. Instead, it functions as a "projection" operator, transforming solutions produced by masked generation from the relaxed space into feasible solutions.

---

> ### Author Response · Authors · 2025-11-21
> **Response to Reviewer VktH (Part 3/4)**
>
> > **Q4: The hyperparameters (e.g., $K$ and $p$) vary across problem types and sizes, as shown in Tables 12–15. It would be helpful to provide the rationale behind these choices. The results in Figures 2–9 demonstrate that model performance is highly sensitive to these hyperparameters.**
>
>
> Thanks for the question. The hyperparameter settings are indeed tuned to each problem, and this is because different tasks have different properties and requirements.
>
> While our method's high-level framework is general, the specifics of how it's applied must adapt to each problem. For example, the impact of masking parts of a path in a problem like TSP is fundamentally different from masking certain nodes in a set cover problem. These differences require tailored hyperparameter settings to ensure the model learns effectively. Using a uniform configuration across all problems would result in suboptimal performance.
>
> It is worth noting again that the cost of selecting an inference configuration is negligible compared to the training time. **For example, training TSP-500 takes approximately 22 hours, whereas selecting the inference configuration requires only 3.6 minutes. The hyperparameter selection is a one-time, automatic post-training step taking <0.3% of total training time.** Even when the post-training overhead is included in the total training time, MaskCO still exhibits significantly shorter training time compared to state-of-the-art methods, as demonstrated in Table 23.
>
> The rationale behind these choices is discussed in Appendix H.2. Besides **semantic interpretation**, we also discuss several additional **practical factors** influencing hyperparameter selection, including data quality and imbalanced learning.
>
> For your convenience, we also include the rationale of semantic intepretation here:
>
> Regarding the two key hyperparameters, $p$ and $K$, each plays a distinct role in guiding the inference process. The parameter $p$ explicitly controls the trade-off between exploitation and exploration: if $p$ is too large, the reconstructed solution remains overly close to the original, potentially causing the correction process to stagnate; if $p$ is too small, the reconstruction may fallback to generating solutions from scratch. Meanwhile, $K$ governs how forward passes are distributed---favoring either more sampling steps per iteration or more iterations with fewer steps. For simpler tasks, smaller values of $K$ are preferable, promoting more iterative refinement. In contrast, for harder tasks where predictions exhibit higher uncertainty, larger $K$ values are beneficial to gather more informative samples.
>
>
>
>
>
>
>
> > **Q5: Some equations and definitions in Sections 3 and 4 could be simplified; currently, they are unnecessarily complicated, which reduces the paper's readability.**
>
>
> Thank you for your insightful comment, and we sincerely apologize for any difficulty our original presentation may have caused. In response, we have simplified the relevant equations and definitions in Sections 3 and 4 to improve clarity and readability. All revised portions are highlighted in blue for easy reference. Our simplifications follow three main principles:
>
> 1. **Streamlined notation for solution vectors**:
>    Instead of using the support operator (e.g., $\mathrm{supp}(\cdot)$) to express inclusion relationships between decision variables, we now directly compare binary vectors using component-wise inequalities such as $\mathbf{x}_1 \leq \mathbf{x}_2$. This change eliminates unnecessary abstraction and makes the relationship more intuitive.
>
> 2. **Refined definition of the Selection Function (Definition 2)**:
>    We have rewritten Definition 2 to retain only its essential mathematical properties while removing redundant or verbose phrasing, resulting in a cleaner and more precise formulation.
>
> 3. **Simplified algorithmic notation (Algorithm 1)**:
>    The function `MultiStepDecoding` was previously called with a long list of parameters, which hindered readability. We now denote it as $\text{Decode}_{f_G,\theta,K}(\cdot)$, moving fixed hyperparameters ($f_G$, $\theta$, $K$) into the subscript and keeping only the dynamically varying arguments in the parentheses. This notational shift significantly reduces visual clutter and enhances comprehension.
>
> We believe these revisions improve the paper's accessibility without compromising technical rigor.

---

> ### Author Response · Authors · 2025-11-21
> **Response to Reviewer VktH (Part 4/4)**
>
> > **Q6: How is MaskCO scalable? Any designs and empirical results to support this claim?**
>
> Thank you for your valuable questions.
>
> For **empirical results**, MaskCO largely benefits from both **scaling model and scaling data**.
>
> **Scaling Model Capacity.** By maintaining the same number of layers, we increase the embedding dimension from 256 to 512, resulting in approximately a 4-fold increase in the number of parameters.
>
> **TSP-1000**
> |        | dim256                | dim512                 |
> | ------ | --------------------- | ---------------------- |
> | T=320  | 23.120, 0.0071%, 18s  | 23.119, 0.0034%, 29s   |
> | T=640  | 23.119, 0.0051%, 33s  | 23.119, 0.0023%, 57s   |
> | T=1280 | 23.119, 0.0038%, 1m6s | 23.118, 0.0013%, 1m52s |
> | T=2560 | 23.119, 0.0027%, 2m8s | 23.118, 0.0008%, 3m43s |
>
> The results demonstrate that scaling the model leads to a more than **3** times reduction in the gap, while also pushing the Pareto frontier of the gap-time tradeoff.
>
>
> **Scaling Training Data.** Experimental results indicate that MaskCO's performance improves with increased data.
>
> **TSP-500 (T=640)**
> | Dataset Size | Gap     |
> | ------------ | ------- |
> | 32k          | 0.0035% |
> | 64k          | 0.0027% |
> | 128k         | 0.0014% |
>
>
> **CVRP-500 (T=640)**
> | Dataset Size | Gap    |
> | ------------ | ------ |
> | 50k          | 1.854% |
> | 100k         | 1.190% |
> | 200k         | 0.813% |
>
>
>
>
>
>
> For **designs**, Transformer-based architecture adopted by MaskCO is inherently scalable. Transformers are well supported by modern deep-learning infrastructure, allowing efficient training and inference on large models and datasets. This design choice makes MaskCO naturally compatible with established scaling practices in CV and NLP, enabling straightforward improvements through increased model capacity and larger training corpora.
>
>
> ---
>
> We hope this response could help address your concerns. We believe that this work contributes to this community with its strong empirical performance. We would sincerely appreciate it if you could reconsider your rating and we are more than happy to address any further concerns you may have.
>
>
>
>
>
> ---------------
>
> [1] Solving Diverse Combinatorial Optimization Problems with a Unified Model.
>
> [2] Learning to Handle Complex Constraints for Vehicle Routing Problems. NeurIPS 2024.

---

### Official Review · Reviewer_Sw1x · 2025-11-03

**Soundness:** 3
**Presentation:** 4
**Contribution:** 3
**Rating:** 8
**Confidence:** 3

**Summary:**

This paper introduces MaskCO, a novel and compelling paradigm for Neural Combinatorial Optimization (NCO) inspired by the success of self-supervised masked auto-encoding in natural language processing and computer vision. The paper identifies a key limitation in
existing NCO methods: they typically treat solutions as monolithic objects during construction or improvement, which is data-inefficient and fails to utilise the local substructures embedded within high quality solutions.

To address this, MaskCO reframes the learning problem as solution-level self-supervised
learning. The core of the training methodology involves strategically masking portions of
known optimal or near-optimal solutions and training a model to reconstruct the missing
components. This approach improves data efficiency by transforming a single (instance,
solution) pair into a vast number of (instance, partial solution) training examples.This process compels the model to internalize fine-grained, localized decision patterns.

For inference, the paper proposes a "mask-and-reconstruct" iterative refinement procedure.
This algorithm begins with an initial solution and progressively improves it by repeatedly
masking a random subset of decision variables and using the trained model to regenerate
them in a single forward pass. This process effectively mimics a highly efficient, parallelized
local search.


The authors validate their approach through extensive experiments on three CO problems: the
Traveling Salesman Problem (TSP), the Capacitated Vehicle Routing Problem (CVRP), and the
Maximum Independent Set (MIS). The results demonstrate that MaskCO achieves new
state-of-the-art performance, significantly outperforming prior neural solvers. The paper also shows that the learned representations are highly versatile, transferring effectively to alternative decoding methods and enabling a powerful self-training paradigm that works even without access to optimal solutions.

**Strengths:**

Novel & Data-Efficient Method: It introduces a new learning paradigm by reframing optimization as a "masked reconstruction" task. This is highly data-efficient, as one optimal solution can be used to create a large number of training examples, forcing the model to learn robust local patterns.

State-of-the-Art Performance & Speed: The model achieves high quality results on TSP, CVRP, and MIS. For example, on TSP-1000, it shows its 9x faster than the previous best neural solver, making its "mask-and-reconstruct" inference a highly efficient, parallelized local search.

High-Quality, Versatile Representations: The learned representations are strong that the model can outperform other methods even when using their decoders. Furthermore, it enables a powerful "optimal-solution-free" mode where the model can teach itself, bootstrapping from weak solutions to high performance.

Significant quality improvements are shown on benchmark datasets such as TSPLIB,

**Weaknesses:**

Check questions

**Questions:**

1. What were the parameters used for the baselines? Were they default parameters from the existing papers or tuned for the target task, Eg:- for BQ-NCO Drakulic et al.  Request the authors to clarify this to ensure fairness of the setup. Were the number of training samples and the training samples used same for baselines and proposed method. A discussion on this would clarify this.

---

> ### Author Response · Authors · 2025-11-21
> **Response to Reviewer Sw1x**
>
> Thank you so much for taking the time to review our submission and for your enthusiastic endorsement. We are truly delighted that you found our paper to be good and recognized the efficiency and versatility of our approach! Your positive feedback is greatly appreciated.
>
>
>
> ---
>
>
> > **Q1: What were the parameters used for the baselines? Were they default parameters from the existing papers or tuned for the target task, Eg: for BQ-NCO (Drakulic et al). Request the authors to clarify this to ensure fairness of the setup.**
>
> Thank you for your valuable feedback. We clarify the baseline settings as follows:
>
> 1) To ensure a fair and accurate comparison with the original papers, we obtained results for nearly all baselines (except SIT [1], which we retrained) by evaluating the publicly released checkpoints from the respective authors under our identical experimental setup. Inference hyperparameters were strictly adhered to as specified in the original papers. For instance, for BQ-NCO [4], we employed a beam width of 16 and attention over the 250 nearest neighbors (KNNs) for instances of TSP/CVRP with 500 and 1000 nodes.
>
>
> 2) Regarding data, some prior methods used different data configurations on CVRP instances with 500 or 1,000 nodes compared to ours. However, methods such as ReLD [2], LEHD [3], and BQ-NCO [4] are inherently designed to generalize across varying problem scales and vehicle capacities using models trained solely on 100-node data, enabling them to be directly evaluated on our dataset with stable performance. In contrast, we observed that SIT exhibited relatively poor generalization on our data, prompting us to retrain it specifically for our setting.
>
> 3) The slight discrepancies between our reported results and those in the original papers (e.g., for BQ-NCO, as noted by the reviewer) stem from differences in the reference solvers used to compute optimality gaps. Specifically, we use HGS as the baseline for gap calculation, whereas some prior works, including BQ-NCO, used LKH. Notably, in the BQ-NCO paper, HGS outperforms LKH by 0.51% on CVRP-100, which aligns with the observed gap differences between the two studies.
>
> We have updated the manuscript accordingly to reflect these clarifications.
>
> > **Q2: Were the number of training samples and the training samples used same for baselines and proposed method. A discussion on this would clarify this.**
>
> Thank you for your insightful question. The number of training samples used in our experiments is detailed in Tables 16, 17, and 18. For your convenience, we summarize the training dataset sizes below:
>
> **TSP**
> |                       | TSP-100 | TSP-500 | TSP-1000 |
> | --------------------- | ------- | ------- | -------- |
> | Training Dataset Size | 1,280K  | 128K    | 128K     |
>
> **CVRP**
> |                       | CVRP-100 | CVRP-500 | CVRP-1000 |
> | --------------------- | -------- | -------- | --------- |
> | Training Dataset Size | 1,536K   | 200K     | 100K      |
>
> **MIS**
> |                       | RB-[200-300] | ER-[700-800] |
> | --------------------- | ------------ | ------------ |
> | Training Dataset Size | 90,000       | 163,840      |
>
>
> ---
>
> We hope this response could help address your concerns. We welcome any further input you may have and remain fully committed to addressing your subsequent feedback and questions.
>
> ----
>
> [1] Boosting neural combinatorial optimization for large-scale vehicle routing problems. ICLR 2025.
>
> [2] Rethinking light decoder-based solvers for vehicle routing problems. ICLR 2025.
>
> [3] Neural combinatorial optimization with heavy decoder: Toward large scale generalization. NeurIPS 2023.
>
> [4] BQ-NCO: Bisimulation Quotienting for Efficient Neural Combinatorial Optimization. NeurIPS 2023.

---

### Author Response · Authors · 2025-12-03
**General Response and Summary (1/3)**

Dear Area Chair,

We sincerely thank the AC and reviewers for their thoughtful and constructive feedback on our paper. Overall, all reviewers (Sw1x, VktH, vnPK, 34zW) recognized the novel paradigm, empirical strength, and potential impact of our approach. We are encouraged that reviewers highlighted MaskCO's **core innovation**: reframing neural combinatorial optimization as solution-level self-supervised learning via masked generation, transforming a single (instance, solution) pair into hundreds of localized training examples and enabling highly efficient, parallelized iterative refinement during inference.

Across reviews, MaskCO is acknowledged for its
- "compelling", "highly data-efficient", "novel", "interesting", "appealing and promising" new learning paradigm by reframing optimization as a "masked reconstruction" task (Sw1x, VktH, vnPK, 34zW);
- "impressive", "state-of-the-art", "significant", "strong" empirical performance "showing 9x faster than the previous best neural solver on TSP-1000" (Sw1x, VktH, 34zW);
- "lightweight" training overhead and "highly efficient, parallelized" inference (Sw1x, 34zW);
- "high-quality, versatile", "strong" representations that "the model can outperform other methods even when using their decoders", "exciting and challenging" topic that "explores a foundational training paradigm that enables effective and scalable representation learning for CO" (Sw1x, 34zW)

Notably, MaskCO achieves **>99% optimality gap reduction** and **up to 10× speedup** over prior state-of-the-art neural solvers on TSP, CVRP, and MIS, setting new standards in both quality and efficiency. And we confirm that **source code will be released upon acceptance**.

---

---

> ### Author Response · Authors · 2025-12-03
> **General Response and Summary (2/3)**
>
> **All concerns raised have been thoroughly addressed in the revision:**
>
> **Major concerns from leading to negative assesments:**
>
> - **Relation to prior masked/self-supervised work** (VktH): We clarify that while both MaskCO and [1] use self-supervised learning, they are fundamentally different in objective, design, and performance. 1）For the algorithmic objective, [1] employs autoregressive next-token prediction for multi-task generalization across CO problems, whereas MaskCO introduces non-sequential, random masked generation to master single tasks with much superior solution quality and speed. 2) Algorithmically, MaskCO aligns with BERT/MAE-style masked modeling, enabling parallel, context-aware reconstruction from arbitrary partial solutions, while [1] follows a GPT-style sequential MDP. 3) Empirically, MaskCO achieves state-of-the-art results (e.g., 0.01% gap on TSP-1000 in 18 seconds), far exceeding the scope of [1], which prioritizes cross-problem transfer over peak performance. The two approaches are complementary but distinct in paradigm and purpose.
>
> - **Supervision dependency** (vnPK): We respectfully emphasize three key points in response to the concern about supervised learning (SL) reliance: 1) **SL remains a vital and highly effective paradigm in neural combinatorial optimization.** Far from being obsolete, SL-based methods continue to provide highly competitive results on problems like TSP, CVRP, and MIS. E.g., MaskCO achieves **>99% optimality gap reduction** and **up to 10× speedup** over prior SOTA neural solvers. Dismissing SL solely due to its use of expert labels overlooks its empirical dominance and ongoing impact in the field. 2) **MaskCO is not limited to expert-supervised settings: it naturally supports self-improvement without any optimal solutions.** Starting with only weak heuristic labels (e.g., 2-opt), our self-training framework iteratively generates high-quality pseudo-labels and refines the model, achieving a **0.000% optimality gap on TSP-100** and SOTA performance **without any expert data**. 3) **We argue that SL and RL are complementary, not opposing.** SL excels at distilling domain knowledge for peak performance on classical problems, while RL enables adaptation to new settings. The most powerful systems, like LLMs, combine both: SL for pretraining, RL for refinement. MaskCO provides a flexible foundation for such synergy, supporting both expert-supervised and fully self-improved training.
>
> - **Additional SOTA baselines** (vnPK): We incorporated comparisons with **PolyNet (ICLR’25)** and **Poppy (NeurIPS’23)**. MaskCO demonstrates clear advantages over these SOTA baselines, including the recent PolyNet presented at ICLR 2025. For the TSP-100 problem, MaskCO achieves a 0.000% Gap with an inspected speedup of approximately 150x compared to PolyNet (EAS). On CVRP-100, MaskCO (T=640) achieves a gap of 0.232%, which is less than half the gap of PolyNet (sampling) (0.496%), while operating within a comparable inference time (32s vs. 27.6s). When given a higher computational budget (T=2560), MaskCO matches the solution quality of PolyNet (EAS), while remaining over 12x faster.
>
> - **Feasibility and constraint handling** (VktH, 34zW): We confirmed that MaskCO ensures **100% feasibility** on TSP, MIS, and CVRP. For CVRP, capacity constraints are enforced via a penalty-augmented 2-opt procedure that can split overloaded routes using empty subroutes. We further demonstrated adaptability to complex constraints by reporting strong results on **TSPTW**, where MaskCO reduces infeasibility by up to 10 times over SOTA neural methods and even outperforms LKH-3 in solution quality (negative gap).

---

> > ### Author Response · Authors · 2025-12-03
> > **General Response and Summary (3/3)**
> >
> > **Other concerns:**
> >
> >
> > - **Baseline fairness and training setup** (Sw1x): We clarified that nearly all baselines were evaluated using authors' official checkpoints under identical inference settings. Training dataset sizes are explicitly reported (Tables 16–18), and differences in optimality gap computation (e.g., HGS vs. LKH) are explained.
> >
> >
> > - **Hyperparameter sensitivity and scalability** (VktH): We provided a principled rationale for key hyperparameters $(K,p)$ based on exploration-exploitation trade-offs and task difficulty. Empirically, we showed that MaskCO scales effectively with **model size** (4× params → >3× gap reduction on TSP-1000) and **training data** (2× data → ~30% gap reduction on CVRP-500). The Transformer backbone ensures compatibility with modern scaling infrastructure.
> >
> > - **Generality and multi-task applicability** (34zW): We clarified that "foundational paradigm" refers to the core *generation mechanism*, not foundation models per se---but that MaskCO can readily integrate with multi-task architectures (e.g., GOAL, MVMoE). We analyzed problem-specific adaptation challenges and possible relaxation strategies for VRPs, scheduling, and graph problems. Experiments on TSPTW broad applicability.
> >
> > - **Presentation and clarity** (VktH, 34zW): We simplified equations and definitions in Sections 3–4, added **Figure 2 (inference)**, included visualizations of partial solution evolution (Appendix R), and provided full mathematical formulation of the architecture (Appendix Q). The introduction has been revised to be more concrete and less abstract.
> >
> > Thank you again for your time and careful attention for our paper. The feedback from the reviewers has significantly strengthened the paper, and we hope our revisions fully address all concerns.
> >
> > We believe this work makes a meaningful contribution to the community through its strong empirical performance, flexible self-supervised paradigm that enables zero- and few-shot finetuning of different strategies for highly competitive performance, and its potential to inspire a broader shift in how NCO methods are designed and trained.
> >
> > We hope that our submission will be evaluated fairly and holistically based on the complete set of revised materials. We sincerely appreciate your time, thoughtful consideration, and the vital role you play in the review process, especially under these unusual circumstances.
> >
> >
> > Best regards,
> > The Authors
> >
> > ---
> >
> > [1] Solving Diverse Combinatorial Optimization Problems with a Unified Model.

---

### Meta-Review · Area_Chair_yHvU · 2026-01-05

**Summary:**

This paper introduces MaskCO, a novel and compelling paradigm for Neural Combinatorial Optimization (NCO) inspired by the success of self-supervised masked auto-encoding in natural language processing and computer vision. Generally, this paper is well written and may contribute to the direction. Compared to previous state-of-the-art neural solvers, MaskCO achieves remarkable performance improvements, exceeding 99% in optimality gap reduction, along with a 10x speedup on the Travelling Salesman Problem (TSP). So, I tend to accept this paper as a poster.

**Reviewer Concerns:**

The illustration from the authors in General Response and Summary is comprehensive and correct.

**Reviewer Scores:**

N/A

---

### Decision · Program_Chairs · 2026-01-26

Accept (Poster)